# SIZE-GENERALIZABLE RNA STRUCTURE EVALUATION BY EXPLORING HIERARCHICAL GEOMETRIES

**Zongzhao Li**[1,2]  **Jiacheng Cen**[1,2]  **Wenbing Huang**[1,2,*]**Taifeng Wang**[3]  **Le Song**[3]

[1]Gaoling School of Artificial Intelligence, Renmin University of China
[2]Beijing Key Laboratory of Big Data Management and Analysis Methods
[3]BioMap Research
lizongzhao2023@ruc.edu.cn, jiacc.cn@outlook.com, hwenbing@126.com
{taifeng, songle}@biomap.com

## ABSTRACT

Understanding the 3D structure of RNA is essential for deciphering its function and developing RNA-based therapeutics. Geometric Graph Neural Networks (GeoGNNs) that conform to the $E(3)$-symmetry have advanced RNA structure evaluation, a crucial step toward RNA structure prediction. However, existing GeoGNNs are still defective in two aspects: 1. inefficient or incapable of capturing the full geometries of RNA; 2. limited generalization ability when the size of RNA significantly differs between training and test datasets. In this paper, we propose EquiRNA, a novel equivariant GNN model by exploring the three-level hierarchical geometries of RNA. At its core, EquiRNA effectively addresses the size generalization challenge by reusing the representation of nucleotide, the common building block shared across RNAs of varying sizes. Moreover, by adopting a scalarization-based equivariant GNN as the backbone, our model maintains directional information while offering higher computational efficiency compared to existing GeoGNNs. Additionally, we propose a size-insensitive $K$-nearest neighbor sampling strategy to enhance the model's robustness to RNA size shifts. We test our approach on our created benchmark as well as an existing dataset. The results show that our method significantly outperforms other state-of-the-art methods, providing a robust baseline for RNA 3D structure modeling and evaluation.

## 1 INTRODUCTION

RNA, or ribonucleic acid, is a pivotal type of molecules essential to myriad processes within biological organisms. Central to its versatility is RNA's 3D structure, which dictates its function and interaction with other molecules (Serganov et al., 2004; Mortimer et al., 2014). Thus, predicting RNA's 3D structure is imperative for advancing our understanding of its biological roles and potential therapeutic applications. Inspired by the breakthroughs achieved by AlphaFold (Jumper et al., 2021) in the field of protein structure prediction, more and more researchers have started to adopt deep learning (such as Graph Neural Net-

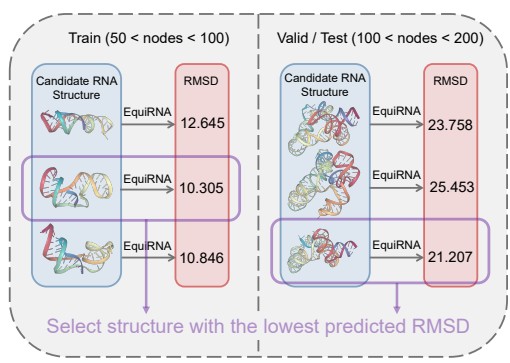

Figure 1: RNA structure evaluation.

works (GNNs)) for RNA 3D structure prediction (Wang et al., 2023a; Shen et al., 2022). Moreover, designing an effective scoring function for evaluation and ranking of RNA structures is a key step in prediction (Townshend et al., 2021; Zhang et al., 2022b), which is the main focus of our paper. The process of RNA structure evaluation can be shown in Fig. 1.

---

*Corresponding author: Wenbing Huang.

RNA structure evaluation presents several challenges: **1.** The predictor we leverage should conform to the E(3) symmetry (3D translation and rotation), as RNA structures are independent to the choice of the coordinate system. **2.** Since large RNA structures are difficult to determine experimentally, currently-known RNA structures are of limited number and mostly of small size (Wang et al., 2023b). As shown in Fig. 2, in the well-known RNAsolo dataset (Adamczyk et al., 2022), among RNA structures with high resolution ($\leq 2.0$ Å), RNAs longer than 100 nucleotides account for only $3.60\%$; while at low resolutions ($> 2.0$ Å), RNAs longer than 100 nucleotides account for $24.62\%$. This observation underscores the current limitations of experimental methods in determining the 3D structures of large RNAs, highlighting a significant challenge in the field of structural biology. Therefore, effectively using smaller RNAs for training a predictor that is generalizable to larger RNAs poses an intriguing research topic, which is identified as the problem of *size generalization* in this paper. As an initial exploration, in this paper, we use smaller RNAs of 50-100 nt for training and larger RNAs of 100-200 nt for testing. The detailed definition of *size generalization* as well as the detailed explanation of the significance for studying this problem are provided in Appendix B.

Several Geometric GNNs (GeoGNNs) have been proposed for RNA structure evaluation (Zhang et al., 2022b; Townshend et al., 2021), but they are still defective in addressing these two challenges. To be specific, PaxNet (Zhang et al., 2022b) utilizes a typical GNN on invariant input (such as bond lengths and angles), making it lose directional information during message passing and thus ineffective in characterising the spatial geometry. ARES (Townshend et al., 2021), on the other hand, employs an equivariant GNN that is able to retain the directional information, but is expanded through high-degree irreducible representations, leading to more computational cost. More importantly, both models are not explicitly and specifically designed to tackle RNAs of varying size, leaving the issue of size generalization less explored.

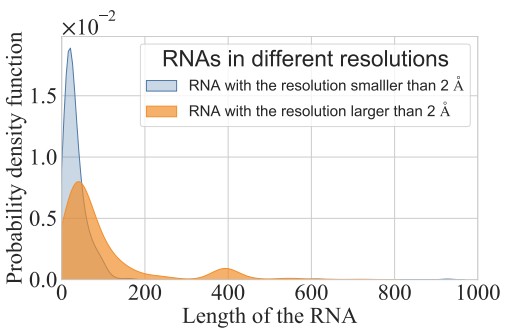

Figure 2: RNA distribution of RNAsolo.

In this paper, we propose a hierarchical equivariant GNN, dubbed as EquiRNA, which consists of three components: the atom-level, subunit-level, and nucleotide-level equivariant message passing processes. Built upon the biological insight that nucleotides are common building blocks shared across RNAs of varying sizes, our model effectively generalizes to different RNA sizes by reusing nucleotides. In particular, we obtain a uniform representation of each nucleotide by undertaking the atom-level and subunit-level message passing over the atoms inside each nucleotide, and then capture the interactions between nucleotides by utilizing the nucleotide-level component. Notably, our model preserves full-atom information throughout all components, in contrast to those pooling-based hierarchical models in other domains (*e.g.* ProNet (Wang et al., 2022)). Moreover, by adopting a scalarization-based equivariant GNN (*i.e.* the EGNN model (Satorras et al., 2021)) as the backbone, our model maintains directional information while offering higher computational efficiency compared to existing GeoGNNs. Additionally, we equip our model with a size-insensitive $K$-nearest neighbor sampling strategy at the nucleotide level. This complements the aforementioned hierarchical structural design, jointly mitigating the challenge of size generalization.

In summary, this paper contributes to the following aspects:

- We introduce a new dataset named **rRNAsolo** for assessing size generalization in RNA structure evaluation. It covers a broader range of RNA sizes, includes more RNA types, and features more recent RNA structures when compared to existing datasets.

- We propose EquiRNA, a hierarchical equivariant graph neural network, tailored to address the size generalization challenge by reusing the representations of nucleotides.

- Extensive experiments on our new dataset rRNAsolo and the existing dataset ARES (Townshend et al., 2021) show that our model achieves better performance across all metrics than SOTA methods, establishing its superiority.

## 2 OUR METHOD

In this section, we first introduce necessary preliminaries related to RNA modeling, and then present the details of our created benchmark rRANsolo for structure evaluation. After this, we describe the architecture of EquiRNA with multi-level equivariant message passing, followed by a size-insensitive $K$-nearest neighbor sampling strategy for handling the imbalance of the size between the training and test RNAs. Fig. 3 shows the whole architecture of our model.

### 2.1 PRELIMINARIES

RNA is a polymeric molecule composed of four types of nucleotides: adenine (A), uracil (U), cytosine (C), and guanine (G). Each nucleotide includes three distinctive chemical sub-units: a five-carbon sugar ribose, a phosphate group, and a nucleobase. The atoms that constitute all components are carbon (C), nitrogen (N), oxygen (O), phosphorus (P), and hydrogen (H). In most cases, different nucleotide shares the same ribose and phosphate group (called ribose-phosphate backbone below), and could contain different type of nucleobase. In this paper, we preserve full-atom representations to enhance the characterization of RNA's 3D structure. In particular, the ribose-phosphate backbone of each nucleotides is denoted as a fully-connected graph $\mathcal{G}_r = (\mathcal{V}_r, \mathcal{E}_r)$. Each node in $\mathcal{V}_r$ encapsulates the information of the $i$-th atom, namely $\vec{v}_{r_i} = (\vec{x}_{r_i}, h_{r_i})$ with $\vec{x}_{r_i} \in \mathbb{R}^3$ being the equivariant 3D coordinate and $h_{r_i} \in \mathbb{R}^c$ being the invariant embedding feature based on atom type. The edges $\mathcal{E}_r$ are constructed by connecting all pairs of atoms in the ribose-phosphate backbone. Similarly, we can define a fully-connected graph for a nucleobase as $\mathcal{G}_b = (\mathcal{V}_b, \mathcal{E}_b)$. To reduce computational complexity, we adopt the strategy outlined in ARES (Townshend et al., 2021) in the above graph construction process, which only considers heavy atoms without hydrogen (H). Finally, a nucleotide is referred to as a joint graph $\mathcal{G} = (\mathcal{V}_r \cup \mathcal{V}_b, \mathcal{E}_r \cup \mathcal{E}_b)$, which is illustrated more clearly in Appendix O.

**Task formulation.** Given a candidate RNA structure with $N$ nucleotides $\{\mathcal{G}_i\}_{i=1}^N$, our goal is to design a neural network $f$ to predict the Root Mean Square Deviation (RMSD) between the candidate and native structures, which will be used to score and rank all candidates to identify the one most closely matching the native structure.

### 2.2 NEW DATASET: RRNASOLO

Although several datasets already exist for RNA structure evaluation (Townshend et al., 2021), they contain RNAs of either limited types or narrow size ranges. Therefore, to extensively explore the problem of size generalization, we devise a novel dataset from the RNAsolo database (Adamczyk et al., 2022), a publicly available online repository that comprises a diverse array of biomolecular information concerning RNA. We call this new dataset as **rRNAsolo**. We provide the details of the dataset construction process as follows.

**Data Collection and Purification.** To better investigate the intrinsic properties of RNA molecules, we exclusively select data from the solo RNA category with a resolution higher than 4 Å. We engage in a detailed cleaning process of the collected data, which is unfolded through two levels. **At the atomic level:** First, we scrutinize all atoms within every RNA molecule to ensure the legitimacy of their valency, eliminating any RNA that contained illicit atoms. Second, we conduct a comprehensive count of atoms within each nucleotide of all RNAs, excluding those with the number of atoms deviated from the normal range. **At the nucleotide level:** First, we analyze the sequencing of nucleotide numbers across all RNAs, discarding those with discontinuous numbering. This discontinuity might be due to insufficient data from solution nuclear magnetic resonance experiments needed to determine their structures. Second, we assess the number of chains contained within the RNA, and exclude any instance with more than two chains, which possibly are redundant chains as a result of experimental techniques. Third, we verify that all RNAs consisted solely of nucleotides and rule out any containing additional residues, such as amino acids. Fourth, aligning with the methodology presented in paper (Townshend et al., 2021), we eliminate RNAs with non-canonical base pairs. Finally, we examine every RNA molecule to reduce conformational ambiguity, specifically for atoms that could occupy possible positions $A$ or $B$. We choose the configuration with higher occurrence frequency, removing all other configurations.

**Data Partitioning and Clustering.** We employ USAlign (Zhang et al., 2022a) to calculate the TM-score between any two RNA samples. USAlign is a metric used to evaluate the structural similarity

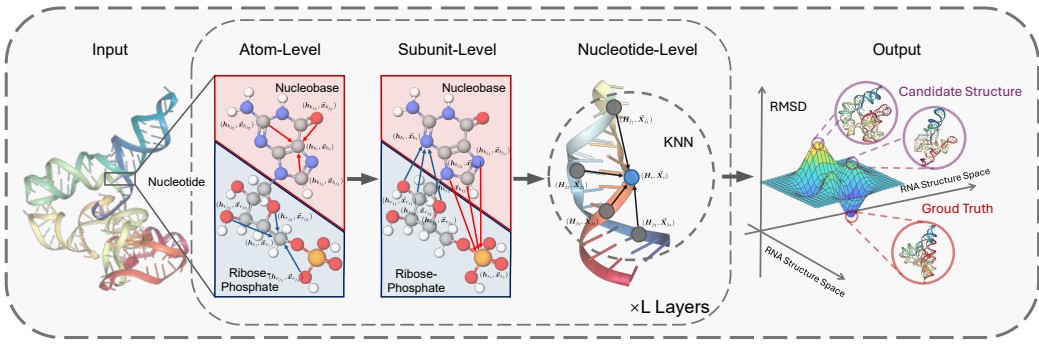

Figure 3: The architecture of EquiRNA. Each layer comprises three levels: atom-level, subunit-level, and nucleotide-level. $(\boldsymbol{h}_r, \vec{\boldsymbol{x}}_r)$ and $(\boldsymbol{h}_b, \vec{\boldsymbol{x}}_b)$ denote the features and coordinates of the atom in ribose-phosphate and nucleobase, respectively. $(\boldsymbol{h}, \vec{\boldsymbol{X}})$ denotes the features and coordinates of each nucleotide. KNN represents our size-insensitive $K$-nearest neighbor sampling strategy. EquiRNA predicts RMSD from native structure and selects structure with the lowest predicted RMSD.

of two RNAs. A TM-score greater than 0.45 indicates that the two RNAs belong to the same RNA family. We then apply the clustering method proposed in USAlign based on the TM-scores to cluster all RNA structures. Specifically, we first sort RNAs by the number of nucleotides in descending order. Then, starting with the largest RNA as the cluster representative, the TM-score is computed with the rest using the USAlign algorithm. If the score is higher than 0.45, it's categorized into the same cluster as the largest RNA; otherwise, the largest RNA forms its own cluster. Repeat the process with the second largest RNA until all RNAs are clustered. Given that RNAs with over 200 nucleotides require considerable time for candidate conformation generation, our validation and test sets primarily focus on RNAs with 100-200 nucleotides, using a training set composed of RNAs with 50-100 nucleotides. Since the number of RNAs within the 100-200 range is smaller than those within 50-100, we first create the validation and test sets by enumerating all clusters and randomly sampling RNAs within the 100-200 range, ensuring maximum coverage of all potential clusters. After the completion of the validation and test sets, we remove the clusters that contain any sample in the validation or test set, and construct the training set using the remaining clusters to prevent data leakage. To be specific, we enumerate the remaining clusters and randomly sample a certain number of RNAs from each cluster to form the training set.

Table 1: The detailed comparisons between our rRNAsolo and ARES.

| Method | Range of sizes | | | Numbers | | | Release date |
|---|---|---|---|---|---|---|---|
| | Train | Valid | Test | Train | Valid | Test | |
| ARES | (10, 50) | (10, 50) | (20, 150) | 18 | 4 | 16 | ~2020 |
| rRNAsolo | (50, 100) | (100, 200) | (100, 200) | 200 | 15 | 15 | ~2023 |

Table 1 compares the statistic of rRNAsolo against the existing benchmark ARES (Townshend et al., 2021). Our **rRNAsolo** exhibits the following advantages: **1.** Wider range of RNA sizes: the training set includes RNAs of 50-100 nucleotides , whereas ARES covers only RNAs less than 50 nucleotides. Our validation and test set consists of RNAs of 100-200 nucleotides, while ARES has only three entities over 100 nucleotides. **2.** A greater number of RNAs: our dataset comprises RNAs approximately 7 times more than ARES. **3.** More recently released data: our dataset contains all the eligible RNA structures up to the latest release, as opposed to ARES which includes RNAs published between 1994-2006 for the training set. Overall, our dataset possesses higher quality and thus enables a more comprehensive evaluation of the current methods. More detailed information about our rRNAsolo can be found in Appendix D.

### 2.3 THE PROPOSED EQUIRNA MODEL

EquiRNA consists of the three-level $E(3)$-equivariant processes: atom-level, subunit-level and nucleotide-level. In this way, EquiRNA largely reuses the common building block (*i.e.* the nucleotide) for RNAs of different sizes, better addressing size generalization. Specifically, the atom-

level and subunit-level modules are able to obtain a uniform representation of each nucleotide, while the nucleotide-level message passing can capture the interaction between different nucleotides.

**(1) Atom-Level Equivariant Message Passing.**

Previous works like (Zhang et al., 2022b) initially calculate E(3)-invariant scalars from atomic coordinates, such as planar angles, dihedral angles, and bond lengths, and then encode and propagate these invariant information through a typical GNN. However, the explicit calculations of scalars suffer from suboptimal computational efficiency and insufficient representational capability. Here, we employ an equivariant GNN to process the atomic coordinates straightforwardly, which is able to retain the orientation information in each layer by propagating geometric information beyond local neighborhoods (Joshi et al., 2023), thus enhancing the modeling of RNAs.

Our model is built upon a prevailing equivariant GNN model, *i.e.*, EGNN (Satorras et al., 2021), by further drawing inspiration from the biological properties of RNA. In particular, for each nucleotide, we conduct atom-level equivariant message passing over the atoms within the ribose-phosphate backbone $\mathcal{G}_r$ and the nucleobase $\mathcal{G}_b$, separately. We denote these two processes as Equivariant Ribose-phosphate Network (ERN) and Equivariant nucleoBase Network (EBN).

**ERN.** For a ribose-phosphate graph $\mathcal{G}_r^l$ comprising $N_r$ nodes (atoms) denoted as $\{\vec{\boldsymbol{x}}_{r_i}^l, \boldsymbol{h}_{r_i}^l\}_{i=1}^{N_r}$ at the $l$-th layer, the formula for updating each node within the graph is given by:

$$\boldsymbol{m}_{r_i r_j}^l = \varphi_m\left(\texttt{rbf}(\|\vec{\boldsymbol{x}}_{r_i}^l - \vec{\boldsymbol{x}}_{r_j}^l\|), \boldsymbol{h}_{r_i}^l, \boldsymbol{h}_{r_j}^l\right),$$
$$\vec{\boldsymbol{x}}_{r_i}^{l+1} = \vec{\boldsymbol{x}}_{r_i}^l + \frac{1}{N_r - 1}\sum_{j \neq i}(\vec{\boldsymbol{x}}_{r_i}^l - \vec{\boldsymbol{x}}_{r_j}^l)\varphi_x(\boldsymbol{m}_{r_i r_j}^l), \tag{1}$$
$$\boldsymbol{h}_{r_i}^{l+1} = \boldsymbol{h}_{r_i}^l + \varphi_h(\boldsymbol{h}_{r_i}^l, \sum_{j \neq i}\boldsymbol{m}_{r_i r_j}^l),$$

where $\varphi_m$, $\varphi_x$ and $\varphi_h$ represent different Multi-Layer Perceptrons (MLPs), $\texttt{rbf}(\cdot)$ is the Radial basis function on the relative distance $\|\vec{\boldsymbol{x}}_{r_i}^l - \vec{\boldsymbol{x}}_{r_j}^l\|$, and $\boldsymbol{m}_{r_i r_j}^l$ is the invariant message from node $j$ to $i$. To derive $\vec{\boldsymbol{x}}_{r_i}^{l+1}$, we first multiply the 1D invariant scalar $\varphi_x(\boldsymbol{m}_{r_i r_j}^l) \in \mathbb{R}$ with the relative coordinate $(\vec{\boldsymbol{x}}_{r_i}^l - \vec{\boldsymbol{x}}_{r_i}^l)$ to recover the orientation information and maintain orthogonality equivariance, and then aggregate all such terms from all neighbors, followed by an addition of the original coordinate $\vec{\boldsymbol{x}}_{r_i}^l$ to ensure translation equivariance. The update of the hidden embedding $\boldsymbol{h}_{r_i}^{l+1}$ follows the conventional message passing strategy. For simplicity, we abstract Eq. (1) as $\mathcal{G}_r^{l+1} = \texttt{ERN}(\mathcal{G}_r^l)$.

**EBN.** Given a nucleobase graph $\mathcal{G}_b^l = \{\vec{\boldsymbol{x}}_{b_i}^l, \boldsymbol{h}_{b_i}^l\}_{i=1}^{N_b}$ at the $l$-th layer, the updates of all atoms are similar to $\texttt{ERN}$ as detailed in Eq. (1). We denote this message passing process as $\mathcal{G}_b^{l+1} = \texttt{ERN}(\mathcal{G}_b^l)$.

**(2) Subunit-Level Equivariant Massage Passing.**

The atom-level message passing above only models the geometry of each subunit, without characterizing their interactions. Biologically, interactions between the ribose-phosphate backbone and the nucleobase involve a complex interplay of forces, e.g., covalent bonds, hydrogen bonds, and so on. To involve interactions, we develop subunit-level equivariant message passing across the atoms in the ribose-phosphate backbone and the nucleobase. We first compute the center of $\mathcal{V}_r$ and $\mathcal{V}_b$, as $\vec{\boldsymbol{x}}_{r_c}^{l+1} = \frac{1}{N_r}\sum_{i=1}^{N_r}\vec{\boldsymbol{x}}_{r_i}^{l+1}$ and $\vec{\boldsymbol{x}}_{b_c}^{l+1} = \frac{1}{N_b}\sum_{i=1}^{N_b}\vec{\boldsymbol{x}}_{b_i}^{l+1}$. Subsequently, we achieve translation invariance by center normalization as $\tilde{\boldsymbol{x}}_{r_i}^{l+1} = \vec{\boldsymbol{x}}_{r_i}^{l+1} - \vec{\boldsymbol{x}}_{r_c}^{l+1}$ and $\tilde{\boldsymbol{x}}_{b_i}^{l+1} = \vec{\boldsymbol{x}}_{b_i}^{l+1} - \vec{\boldsymbol{x}}_{b_c}^{l+1}$, respectively. Then, the update process for each atom in $\mathcal{G}_r$ is given by:

$$\left.\begin{aligned}\boldsymbol{q}_i &= \varphi_q\left(\boldsymbol{h}_{r_i}^{l+1}, \texttt{rbf}(\|\tilde{\boldsymbol{x}}_{r_i}^{l+1}\|)\right) \\ \boldsymbol{k}_j &= \varphi_k\left(\boldsymbol{h}_{b_j}^{l+1}, \texttt{rbf}(\|\tilde{\boldsymbol{x}}_{b_j}^{l+1}\|)\right) \\ \boldsymbol{v}_j &= \varphi_v\left(\boldsymbol{h}_{b_j}^{l+1}, \texttt{rbf}(\|\tilde{\boldsymbol{x}}_{b_j}^{l+1}\|)\right)\end{aligned}\right\} \implies \left\{\begin{aligned}\alpha_{ij} &= \texttt{Softmax}(\boldsymbol{q}_i^\top \boldsymbol{k}_j) \\ \vec{\boldsymbol{x}}_{r_i}^{l+2} &= \vec{\boldsymbol{x}}_{r_i}^{l+1} + \frac{1}{N_b}\sum_{j \in \mathcal{V}_b}\alpha_{ij}\tilde{\boldsymbol{x}}_{b_j}^{l+1}\varphi_{xr}(\boldsymbol{v}_j) \\ \boldsymbol{h}_{r_i}^{l+2} &= \boldsymbol{h}_{r_i}^{l+1} + \varphi_{hr}\left(\boldsymbol{h}_{r_i}^{l+1}, \sum_{j \in \mathcal{V}_b}\alpha_{ij}\boldsymbol{v}_j\right)\end{aligned}\right. \tag{2}$$

where $\varphi_q$, $\varphi_k$, $\varphi_v$, $\varphi_{xr}$ and $\varphi_{hr}$ are all MLPs, $\texttt{rbf}(\cdot)$ is the Radial basis function. We apply the attention mechanism to depict the interaction between each atom in $\mathcal{G}_r$ and $\mathcal{G}_b$. The invariant quantities $\boldsymbol{q}_i$, $\boldsymbol{k}_j$ and $\boldsymbol{v}_j$ refer to the query, key and value features, respectively, and $\alpha_{ij}$ signifies the

attention weights, with higher values indicating more pronounced interactions between atom $i$ and $j$. The aforementioned procedure can be succinctly expressed as $\mathcal{G}_r^{l+2} = \texttt{EB2R}(\mathcal{G}_r^{l+1}, \mathcal{G}_b^{l+1})$.

Following this, we employ an analogous approach to model the interactions exerted by the ribose-phosphate backbone on the nucleobase, leading to $\mathcal{G}_b^{l+2} = \texttt{ER2B}(\mathcal{G}_b^{l+1}, \mathcal{G}_r^{l+1})$.

### (3) Nucleotide-Level Equivariant Message Passing.

Upon completing the atom-level and subunit-level modeling, we advance to message passing at the nucleotide level. We aggregate the features of all atoms within the the same nucleotide, which have been updated as described previously, deriving the nucleotide feature by summation: $\boldsymbol{h}^{l+2} = \sum_{i=1}^{N_r} \boldsymbol{h}_{r_i}^{l+2} + \sum_{j=1}^{N_b} \boldsymbol{h}_{b_j}^{l+2}$, and the coordinate matrix by concatenating all atomic coordinates: $\vec{\boldsymbol{X}}^{l+2} \in \mathbb{R}^{3 \times (N_r + N_b)}$. To distinguish between nucleotides, we denote different nucleotide via subscript $i$: $(\vec{\boldsymbol{X}}_i^{l+2}, \boldsymbol{h}_i^{l+2})$ with $N_i$ being the number of atoms in this nucleotide.

It is challenging to design a function on $\vec{\boldsymbol{X}}_i^{l+2}$, since different nucleobase could have a distinct number of atoms, implying that the dimension of $\vec{\boldsymbol{X}}_i^{l+2}$ varies. Here, we address this challenge by using templates. First, we define 4 atom templates $\{\boldsymbol{W}_k^a \in \mathbb{R}^{d_a}\}_{k=1}^4$, as there are 4 types of atoms used in our model. Each atom template is learnable with the number of channels as $d_a$. The atom-level templates are engineered to capture the intrinsic characteristics of all atoms. We further define nucleotide templates $\{\boldsymbol{W}_k^n \in \mathbb{R}^{N_i \times d_n}\}_{k=1}^4$, where 4 reflects the 4 types of nucleotides, and $d_n$ reflects the dimension of the nucleotide template. The nucleotide-level templates aim to capture the unique attributes of the same atom when situated in different categories of nucleotides. In summary, $\boldsymbol{W}_k^a$ and $\boldsymbol{W}_k^n$ are two learnable templates that store information about each type of atom and nucleotide, respectively. In $\boldsymbol{W}_k^n$, different row represents the characteristics of different atom, and therefore does not share the same representation. We apply these templates to $\vec{\boldsymbol{X}}_i^{l+2}$ to derive fixed-size message from nucleotide $j$ and $i$.

In detail, we begin by calculating the distances between any pair of atoms in nucleotide $i$ and $j$, yielding the distance matrix $\boldsymbol{D}_{ij} \in \mathbb{R}^{N_i \times N_j}$. We then compute the center of each nucleotide, denoted as $\vec{\boldsymbol{x}}_c^{l+2} = \frac{1}{N_r + N_b}(\sum_{i=1}^{N_r} \vec{\boldsymbol{x}}_{r_i}^{l+2} + \sum_{j=1}^{N_b} \vec{\boldsymbol{x}}_{b_j}^{l+2})$. The coordinates and features of each nucleotide are updated as follows:

$$
\begin{aligned}
\boldsymbol{m}_{ij}^n &= \left(W_{\boldsymbol{t}_i}^a \oplus W_{\tau_i}^n\right)^\top \left((\boldsymbol{p}_{\tau_i}\boldsymbol{p}_{\tau_j}^\top) \odot \boldsymbol{D}_{ij}\right) \left(W_{\boldsymbol{t}_j}^a \oplus W_{\tau_j}^n\right), \\
\boldsymbol{m}_{ij}^{l+3} &= \varphi_{hn}\left(\boldsymbol{h}_i, \boldsymbol{h}_j, \frac{\boldsymbol{m}_{ij}^n}{\|\boldsymbol{m}_{ij}^n\|_{\mathrm{F}}}\right), \\
\vec{\boldsymbol{X}}_i^{l+3} &= \vec{\boldsymbol{X}}_i^{l+2} + \frac{1}{|\mathcal{N}(i)|} \sum_{j \in \mathcal{N}(i)} \texttt{Pool}\left(\varphi_{xn}\left(\boldsymbol{m}_{ij}^{l+3}\right)\right) \cdot \left(\vec{\boldsymbol{X}}_i^{l+2} - \vec{\boldsymbol{x}}_{c_j}^{l+2}\right), \\
\boldsymbol{h}_i^{l+3} &= \boldsymbol{h}_i^{l+2} + \varphi_{hn}\left(\boldsymbol{h}_i^{l+2}, \sum_{j \in \mathcal{N}(i)} \boldsymbol{m}_{ij}^{l+3}\right),
\end{aligned}
\tag{3}
$$

where $\vec{\boldsymbol{x}}_{c_j}^{l+2}$ represents the center of nucleotide $j$, $\oplus$ represents concatenations along the channel dimension; $\boldsymbol{t}_i \in \{1, \ldots, 4\}^{N_i}$ are vectors concatenated by all atom types in nucleotide $i$ and $\tau_i \in \{1, \ldots, 4\}$ represents its nucleotide type; the dimension of $W_{\boldsymbol{t}_i}^a$ and $W_{\tau_i}^n$ are $\mathbb{R}^{N_i \times d_a}$ and $\mathbb{R}^{N_i \times d_n}$, respectively; $\boldsymbol{p}_{\tau_i}$ and $\boldsymbol{p}_{\tau_j}$ are two learnable vectors used to model the atom-wise correlation in the distance matrix $D_{ij}$; $\mathcal{N}(i)$ denotes all neighboring nucleotides of nucleotide $i$. The dimension of the derived message $\boldsymbol{m}_{ij}^n$ in the first equation keeps unchanged regardless of the value of $N_i$. The message $\boldsymbol{m}_{ij}^n$ is a matrix with dimension $\mathbb{R}^{(d_a+d_n) \times (d_a+d_n)}$. By default, we will flatten $\boldsymbol{m}_{ij}^n$ into a vector of size $\mathbb{R}^{(d_a+d_n)^2}$ before feeding it to the update function $\varphi_{hn}$. The pooling operation $\texttt{Pool}(\cdot)$ with adaptive stride ensures the output of $\varphi_{xn}(\boldsymbol{m}_{ij}^{l+3})$ to have the same dimension as $\vec{\boldsymbol{X}}_i^{l+2}$.

**Size-insensitive $K$-nearest neighbor sampling strategy.** To further alleviate the size-generalization issue, we propose a size-insensitive $K$-nearest neighbor sampling strategy, designed to leverage the abundant smaller RNAs for training and enable the model to generalize effectively to larger RNAs. In detail, we first perform $M$-nearest neighbor sampling for all RNAs, then randomly select $K$ out of these $M$ neighbors ($K < M$) with equal probability. Consequently, the model has been exposed to $M$-neighbors' information during training, allowing it to transfer the spatial information of $M$-neighbors learned from the training set to larger RNAs in the test set. This process is

analogous to randomly masking nodes and dropping edges in the graph during each training period, which renders the model less sensitive to variations in the size of RNAs and enhances its robustness.

**Learning objective.** After implementing $L$ layers of computations using the modules described in § 2.3, we perform the final calculation on the output $\boldsymbol{h}_i^{l+3}$ from Eq. (3) to predict the RMSD between the candidate structure and the native structure. Specifically, we first sum up the features of all nucleotides, and then feed them through a MLP to predict a scalar. This scalar represents the predicted RMSD by the model. Ultimately, we employ Smooth $\ell_1$ loss as our loss function. This process is delineated as follows:

$$\mathcal{L} = \mathcal{L}_{\texttt{smooth\_l1}} \left( y, \; \texttt{MLP} \left( \sum_{i=1}^{N_{\text{nu}}} \boldsymbol{h}_i^{l+3} \right) \right), \tag{4}$$

where $y$ is a scalar, calculated by FARFAR2 when generating the candidate structure, representing the ground truth label. $N_{\text{nu}}$ represents the number of nucleotide in RNA. A pivotal characteristic of our model is its $\text{E}(3)$-equivariance, with proofs provided in Appendix C.

## 3 EXPERIMENT

**Datasets.** We conducted exhaustive experiments on two datasets comprising a wide range of RNA structures. **1. rRNAsolo:** In our meticulously designed dataset, we employs candidate structures of RNAs with 50-100 nt as training set and candidate structures with 100-200 nt as validation and test sets. The dataset rRNAsolo consists of 80k/6k/6k candidate structures generated from 200/15/15 RNAs for training, validation, and test sets, respectively. **2. ARES:** Within ARES dataset (Townshend et al., 2021), 14 RNAs with 19-47 nt form training set and 4 RNAs with 17-41 nt form validation set, each generating 1k candidate structures. Additionally, 16 RNAs (only three of which are larger than 100 nt) with 27-147 nt form test set, with each RNA generating 5k candidate structures. The training, validation, and test sets include 14k/4k/80k candidate structures, respectively.

**Baselines and Implementation.** We compare our method with the following baselines: the invariant GNNs including PaxNet (Zhang et al., 2022b) and RDesign (Tan et al., 2024), and the equivariant GNNs including ARES (Townshend et al., 2021), EGNN (Satorras et al., 2021), dyMEAN (Kong et al., 2023b), and GET (Kong et al., 2024). We use the default configurations in the corresponding source codes for all baselines. Details of the configuration settings are provided in Appendix E and Appendix P.

**Metrics.** For each RNA in the validation and test sets, the models are required to score all related candidate structures and selects the best-scoring one. Subsequently, the RMSD between the selected best structure and the native structure is utilized as the criterion for assessing the model's evaluation ability. The metrics include: **1.Median RMSD**: Following the evaluation metrics in Zhang et al. (2022b); Townshend et al. (2021), we calculate the median RMSD across RNAs; **2.Mean RMSD:** We also calculate the mean RMSD across RNAs to further evaluate the model's scoring capability comprehensively. **3.Relative Error of Median/Mean RMSD:** For each RNA , there is a theoretical minimum RMSD value among all the candidate structures. Thus, we calculate the relative error of the median/mean RMSD with respect to the median/mean of the minimum RMSD.

### 3.1 MAIN RESULTS

**Results on rRNAsolo.** Table 2 provides a detailed comparison of various methods' performance on rRNAsolo. Within this dataset, our model shows significant improvements on all evaluation metrics compared to the current SOTA methods for both validation and test sets. We can observe that: **1.** Our model achieves the greatest performance gains by 2.00 and 1.55 across the Mean RMSD and Medium RMSD metrics on both validation and test sets. Furthermore, as indicated by Relative Error of Mean RMSD and Relative Error of Medium RMSD, our model yields improvements of up to 17% and 13% on the validation and test sets, respectively. Given the small performance differences among SOTA models, this demonstrates the efficiency and superiority of our method in RNA 3D structure modeling. **2.** Analyzing from the perspective of equivariance, the PaxNet, which retains only invariant feature, underperforms equivariant models like dyMEAN and EquiRNA. This underscores the importance of considering physical symmetry in RNA 3D structure modeling. **3.** As a method in protein domain, dyMEAN still exhibits a certain gap in performance compared to EquiRNA. This suggests that despite structural similarities, the intricacy and flexibility of RNA

Table 2: All four metrics on the validation and test sets of rRNAsolo. The four metrics in the table, ER, DR, R-ER, and R-DR, correspond to the abbreviations of the previously mentioned metrics Mean RMSD, Median RMSD, Relative Error of Mean RMSD, and Relative Error of Median RMSD, respectively.

| | Validation Set | | | | Test Set | | | |
|---|---|---|---|---|---|---|---|---|
| | ER ↓ | DR ↓ | R-ER ↓ | R-DR ↓ | ER ↓ | DR ↓ | R-ER ↓ | R-DR ↓ |
| ARES (Townshend et al., 2021) | 22.13 | 22.45 | 1.12 | 1.22 | 20.98 | 20.89 | 0.84 | 0.84 |
| EGNN (Satorras et al., 2021) | 22.12 | 23.12 | 1.01 | 1.34 | 20.99 | 20.42 | 0.71 | 0.69 |
| PaxNet (Zhang et al., 2022b) | 22.71 | 24.00 | 1.12 | 1.43 | 21.64 | 21.53 | 0.76 | 0.78 |
| dyMEAN (Kong et al., 2023b) | 21.34 | 20.94 | 0.99 | 1.11 | 20.91 | 20.52 | 0.71 | 0.70 |
| RDegisn (Tan et al., 2024) | 21.99 | 20.79 | 1.05 | 1.11 | 20.22 | 19.34 | 0.65 | 0.60 |
| GET (Kong et al., 2024) | 21.61 | 21.68 | 1.01 | 1.20 | 20.03 | 19.19 | 0.63 | 0.59 |
| EquiRNA | **19.77** | **19.62** | **0.84** | **0.99** | **18.22** | **17.79** | **0.48** | **0.47** |

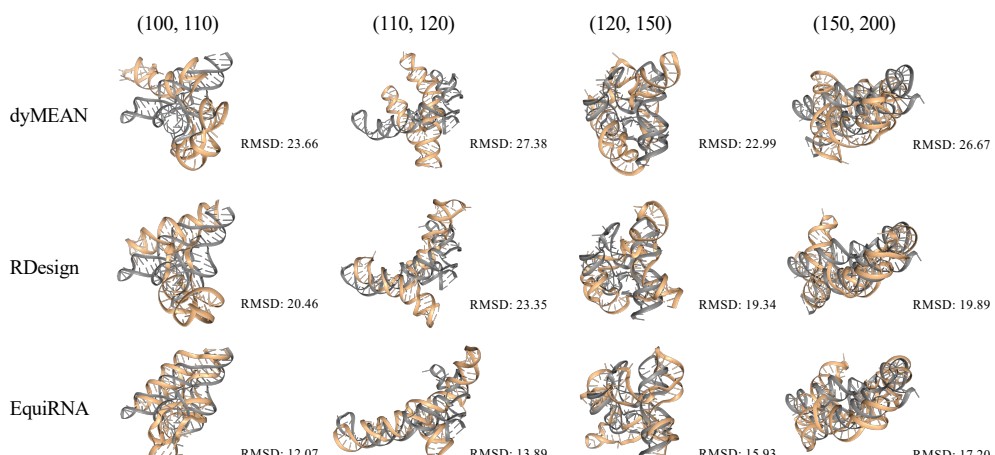

Figure 4: Native structures (gray) versus predicted structures (beige) by EquiRNA and other models.

structures places higher demands on a model's ability to capture RNA characteristics. **4.** Inferior performance is observed of GET, attributable to its lack of specialized mechanisms for addressing the unique structural features of RNAs. This limitation likely stems from GET's general-purpose design, rather than RNA-specific optimization.

To comprehensively compare our model's ability to assess RNAs of various sizes with current SOTA models, Fig. 5 further analyzes each model's evaluation capability across all RNAs in the validation and test sets. We first divide the RNAs into size intervals of 100-110, 110-120, 120-150, and 150-200, based on the distribution of RNA sizes. Then we calculate the average prediction values for RNAs in each interval. Fig. 5 clearly shows our method's superior predictive results across various RNA sizes, indicating our approach effectively addresses the size generalization problem, transferring the knowledge of nucleotide interactions learned from small RNAs to larger RNAs. Moreover, we selecte four RNAs from the size intervals discussed previously. Fig. 4 displays the native structures, compared with the predicted structures of the current SOTA models and our model.

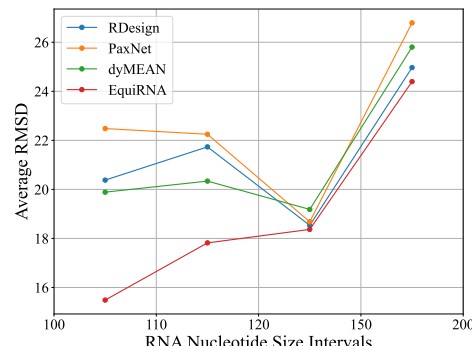

Figure 5: Comparison of different methods in the validation and test sets.

Table 3: Results across four metrics on the test set of ARES dataset.

| Method | ER | DR | R-ER | R-DR |
|--------|------|-------|------|------|
| ARES | 12.33 | 11.22 | 1.02 | 0.90 |
| EGNN | 13.47 | 12.12 | 1.21 | 1.05 |
| PaxNet | 13.15 | 10.28 | 1.16 | 0.74 |
| dyMEAN | 13.56 | 11.30 | 1.23 | 0.91 |
| RDegisn | 13.37 | 12.96 | 1.20 | 1.20 |
| GET | 13.29 | 11.62 | 1.18 | 0.97 |
| EquiRNA | **11.74** | **9.57** | **0.93** | **0.62** |

Table 4: The Relative Ranking of all the models on rRNAsolo and ARES datasets.

| Method | rRNAsolo | ARES |
|--------|----------|-------|
| ARES | 0.591 | 0.294 |
| EGNN | 0.435 | 0.325 |
| PaxNet | 0.628 | 0.343 |
| dyMEAN | 0.603 | 0.347 |
| RDesign | 0.476 | 0.351 |
| GET | 0.423 | 0.343 |
| EquiRNA | **0.280** | **0.238** |

**Results on ARES.** To further assess the versatility and generalization capabilities of our model, we conducted experiments on the public dataset ARES. As shown in Table 3, our method achieves optimal results on all four metrics. Despite the smaller size and fewer training samples in this dataset, our model still outperforms the current SOTA method ARES, demonstrating its capability to encode essential structural information within and between nucleotides. Additionally, following the ARES paper, we present the results of all methods on the rRNAsolo and ARES datasets using the 1 best-scoring model and 10 best-scoring models in Table 9 of Appendix F.1. Morever, we also provide detailed analyses of the experimental results in Appendix F.2.

**High-level RMSD values presented in rRNAsolo.** The RMSD values observed in our rRNAsolo dataset are higher than those in ARES dataset. We briefly elucidate the underlying factors contributing to this discrepancy and introduce a more appropriate metric for evaluating model performance. Comprehensive analyses of this phenomenon are provided in Appendix G. There are primarily two reasons: (1) Our rRNAsolo is more challenging with wider range of RNA sizes and a greater number of large RNAs. (2) The reported RMSD values are predominantly contingent upon the quality of candidate structures; however, for large RNAs, existing predictive methods often struggle to generate high-quality candidates with low RMSD values. To provide a more direct understanding of model's evaluation capability, we have reported the **Relative Ranking** metric of all models on rRNAsolo and ARES datasets in Table 4. The calculation of **Relative Ranking** is as follows: the evaluation model first ranks all candidate structures' ground-truth RMSD values in an ascending order out of $N$ candidate structures in total. Then we identify the order of evaluation model's selected structure as $E$, and Relative Ranking is calculated as *(E/N)*. Clearly, our method significantly outperforms other evaluation models on these two datasets. Moreover, our method shows consistent performance across both rRNAsolo and ARES datasets, while the performance of other methods declines noticeably on rRNAsolo. Compared to metrics we originally used, Relative Ranking provides a more scientific and intuitive assessment of the model's ability to rank candidate structures. One can directly use ranking metric to evaluate whether evaluation model offers reliable guidance for RNA selection.

**Further exploratory experiments.** We also conduct a series of exploratory experiments, the details of which are presented in the appendices, including: the incorporation of molecular chirality in the model architecture (Appendix H); the influence of the noise (Appendix N); the model performance under a more general scenario where size generalization is not a constraint (Appendix J); performance comparisons with other leading models (Appendix L); the performance of the model after fine-tuning on a dataset constructed using the recently published RhoFold (Shen et al., 2024) as a novel candidate structure generator (Appendix Q); and the analysis of some error cases (Appendix M).

**Complexity analyses.** We now proceed to conduct a complexity analysis of the different models. The efficiency analysis and comparison are provided in Table 5. In this table, $N_a$, $n_a$, $K_a$, $K_{nu}$, $C$, and $L$ denote number of atoms in each RNA, number of atoms in each nucleotide, number of nearest atom neighbors, number of nearest nucleotide neighbors, channel size of high-degree tensors, and angular order, respectively. The $K_{a1}$ and $K_{a2}$ in PaxNet denotes the the number of nearest atom neighbors in two different settings, the $d_a$ and $d_{nu}$ in EquiRNA denotes the embedding size of the atom template and the nucleotide template. Our EquiRNA costs much less inference time than the methods that using high-degree irreducible representations (ARES, SE(3)-Transformer (Fuchs et al., 2020), Equiformer (Liao & Smidt, 2022) and SE(3)-Hyena (Moskalev et al., 2024)), and is even faster than EGNN since EquiRNA employs the hierarchical modeling and can fulfil desirable performance with a smaller neighbor size $K$ than EGNN in the atom-level message passing.

Table 5: Complexity and inference time.

| Method | Complexity | Inference Time |
|---|---|---|
| ARES | $\mathcal{O}(N_a K_a C^3 L^6)$ | 0.095s |
| SE(3)-Transformer | $\mathcal{O}(N_a K_a C^3 L^6)$ | - |
| Equiformer | $\mathcal{O}(N_a K_a C^3 L^6)$ | - |
| SE(3)-Hyena | $\mathcal{O}(N_a(K_a + \log N_a)C^3 L^6)$ | - |
| EGNN | $\mathcal{O}(N_a K_a)$ | 0.042s |
| PaxNet | $\mathcal{O}(N_a(K_{a1} + K_{a2}))$ | 0.035s |
| EquiRNA | $\mathcal{O}(n_a N_a + (n_a^2 + n_a(d_a + d_{nu}))K_{nu}N_{nu})$ | 0.038s |

## 3.2 ABLATION STUDIES

Comprehensive ablation studies are carried out to evaluation the impact of each component of the EquiRNA model on performance, with specific results presented in Table 6. The At, Su, Nu, Sa, and Eq correspond to the abbreviations of the atom-level, subunit-level, nucleotide-level, size-insensitive $K$-nearest neighbor sampling strategy, and equivariance, respectively. We have the following observations: **1.** The experiments indicate that eliminating any one of atom-level, subunit-level, or nucleotide-level submodule will lead to a decrease in performance. This phenomenon confirming that EquiRNA effectively encodes critical information of RNA 3D structures, such as dihe-

Table 6: Ablation studies on rRNAsolo.

|  | ER | DR | R-ED | R-DE |
|---|---|---|---|---|
| w/o At | 19.38 | 19.40 | 0.58 | 0.61 |
| w/o Su | 19.53 | 19.57 | 0.59 | 0.62 |
| w/o Nu | 20.25 | 19.41 | 0.65 | 0.61 |
| w/o Sa | 19.40 | 17.94 | 0.58 | 0.49 |
| w/o Eq | 20.53 | 19.42 | 0.67 | 0.61 |
| w/o $\boldsymbol{W}_k^a$ | 20.03 | 19.64 | 0.63 | 0.63 |
| w/o $\boldsymbol{W}_k^n$ | 18.69 | 17.97 | 0.52 | 0.49 |
| EquiRNA | **18.22** | **17.79** | **0.48** | **0.47** |

dral angles, and facilitates message passing within and between these three levels. **2.** The removal of the nucleotide-level results in a significant decline in performance, revealing the significance of nucleotide interactions. **3.** The ablation of the size-insensitive $K$-nearest neighbor sampling strategy also leads to diminished performance, demonstrating that this strategy allows the model to engage with a more extensive range of local neighbors during training, which is beneficial when coping with the large RNAs. **4.** From the results presented in Row 5, the introduction of equivariance enhances the model's ability to encode physical symmetry, resulting in a stronger capacity for 3D structure representation. **5.** Removing atom template $W_k^a$ or nucleotide template $W_k^n$ hinders the performance, indicating their ability to capture the biology information. More ablation studies on subunit-level and nucleotide-level can be found in Appendix I.

## 4 CONCLUSION

We introduce EquiRNA, a hierarchical equivariant model designed for RNA structure evaluation. The innovation of this model lies in deconstructing RNA structures into multiple levels. It performs message passing and aggregation from the bottom-up, starting at the atom-level and progressing through subunit-level and nucleotide-level, while preserving full-atom details. Additionally, we incorporate a size-insensitive $K$-nearest neighbor sampling strategy to mitigate challenges associated with generalizing to large RNAs. The entire architecture of EquiRNA guarantees the critical E(3)-equivariance to properly encode physical symmetries and enhance its expressive capabilities. Comprehensive experiments on two datasets with a broad range of RNA sizes demonstrate that the selected structures of our model have significantly lower RMSD values, showcasing its superior performance. We hope that EquiRNA will serve as a robust baseline for the field and foster further advancement.

## ACKNOWLEDGMENTS

This work was jointly supported by the following projects: the National Natural Science Foundation of China (No. 62376276); Beijing Nova Program (No. 20230484278); Beijing Outstanding Young Scientist Program (No. BJJWZYJH012019100020098), the Fundamental Research Funds for the Central Universities, and the Research Funds of Renmin University of China (23XNKJ19). We thank Jiahui Su for her insightful suggestions that improved this paper.

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

## A    RELATED WORK

**RNA Structure Prediction and Evaluation.** Due to the scarcity of RNA structural data and the inherent variability of RNA structures, predicting RNA 3D structures has long been a challenge in the academic community. Furthermore, a key subtask of 3D structure prediction, which involves accurately evaluating and scoring the predicted set of RNA 3D candidate conformations, also presents a substantial challenge, largely because many features closely related to RNA structural energy have yet to be fully understood. Traditional approaches often employ physics-based methods (Li & Chen, 2023) or knowledge-based methods (Das & Baker, 2007; Watkins et al., 2020) for structure prediction and leverage statistical potentials (Samudrala & Moult, 1998; Wang et al., 2015) for conformation evaluation. Recently, deep learning-based methodologies have begun to emerge, with DeepFoldRNA (Pearce et al., 2022) and trRosettaRNA (Wang et al., 2023a) employing Transformer networks as their main architecture to predict 3D structures. ARES (Townshend et al., 2021) and PaxNet (Zhang et al., 2022b) modeling and scoring candidate conformations at the atomic level with notable success. Our research, drawing upon RNA's biological nature, introduces a hierarchical modeling approach to enhance the accuracy of structural scoring.

**Geometric GNNs.** In scientific scenarios such as molecular property prediction (Jiao et al., 2023; 2024; Liu et al., 2025), dynamics simulation (Han et al., 2022; Zhang et al., 2024; Wu et al., 2023; Xu et al., 2024b), and protein design (Kong et al., 2023a; Yu et al., 2024; Yue et al., 2025), a large amount of 3D structured data is contained, which is often modeled as *geometric graphs*. In order to efficiently process this type of special data structure, *geometric GNNs* came into being and achieved great success (Han et al., 2024). Different from invariant GNNs (*e.g.* SchNet (Schütt et al., 2018), DimeNet (Gasteiger et al., 2020), DisGNN (Li et al., 2024b), GeoNGNN (Li et al., 2025a), QMP (Yue et al., 2024)) with limited expressiveness (Joshi et al., 2023) and high-degree steerable models (*e.g.* TFN (Thomas et al., 2018), SEGNN (Brandstetter et al., 2021), MACE (Batatia et al., 2022)) which incur a lot of computational overhead on tensor products, scalarization-based models (*e.g.* EGNN (Satorras et al., 2021), GMN (Huang et al., 2022), PAINN (Schütt et al., 2021)) utilize scalars as weights to combine geometric vector information has achieved remarkable results in both accuracy and efficiency. Motivated by this , spherical-scalarization approaches (*e.g.*, SO3KRATES (Frank et al., 2024), HEGNN (Cen et al., 2024), GotenNet (Aykent & Xia, 2025)) employ invariant information—such as inner products or moduli-derived from high-degree representations to strengthen model expressivity. More recently, ETNN (Battiloro et al., 2025) and Equi-LLM (Li et al., 2025b) enhance model capabilities by injecting information obtained through deep topological learning and large language models into the scalar, respectively. In the RNA scoring task, ARES (Townshend et al., 2021) and PaxNet (Zhang et al., 2022b) adopt high-degree steerable GNNs and invariant GNNs, respectively, revealing inherent limitations as described above. In this paper, we introduce EGNN (Satorras et al., 2021) as the backbone that preserves the model's equivariant characteristics while achieving computational efficiency.

**Size Generalization.** In the realm of graph neural networks (GNNs), particularly in biological contexts, the size generalization capability of GNNs remains underexplored. Existing literature has noted models trained on small-scale graphs often fail to retain their performance when generalized to larger graphs (Ji et al., 2022; Chu et al., 2023; Zhou et al., 2022; Chen et al., 2022). In pursuit of a more comprehensive understanding of this phenomenon, Yehudai et al. (2021) identify discrepancies in local structural $d$-patterns as the cause for the reduction in generalization capacity. Buffelli et al. (2022) introduce graph coarsening to simulate size shift, thereby augmenting the model's robustness. Additionally, Yang et al. (2022) advocate for the identification of invariant substructures within graphs, such as molecular substructures, to ascertain key (bio)chemical properties. To solve this problem, our work presents a high-quality dataset specifically curated to validate model generalizability over varying sizes of RNA data. Additionally, we introduce a hierarchical equivariant GNN as well as a size-insensitive $K$-nearest neighbor sampling strategy that enhances model's adaptability to large-scale RNAs.

## B    DEFINITION AND SIGNIFICANCE OF SIZE GENERATION

**The definition of size generation:**

We define the size generalization problem as follows: Given two datasets composed of different RNA structures, one designated as the training set and the other as the test set. The training set and test set have the following two characteristics: 1. The structural similarity (measured by TM-score) between any RNA in the training set and any RNA in the test set is less than 0.45 to prevent data leakage; 2. All RNA sequences in the test set are longer than any RNA sequence in the training set. Our research question is: After training on the training set, how well does the model perform on the test set? We measure the model's generalization ability by evaluating its accuracy in assessing the 3D structures of RNA in the test set.

**The significance of studying size generation:**

During our experiments, we observe that the average length of RNA in the existing ARES dataset (Adamczyk et al., 2022) is significantly shorter during training compared to testing (26.42 nt vs. 78.12 nt). This pattern is not unique to the ARES dataset; similar statistics from other RNA structure datasets indicate a consistent trend: the proportion of large RNAs is much smaller than that of smaller ones. Specifically:

(1) According to statistics from the RCSB PDB (Berman et al., 2000), in the early RNA 3D structure datasets (1978-2010), RNAs longer than 100 nucleotides accounted for only 9.81% of the total, with the remaining being shorter RNAs.

(2) Based on statistics from the RNAsolo dataset (Adamczyk et al., 2022), currently, RNAs longer than 100 nucleotides account for only 16.79% of all experimentally determined RNA 3D structures. In contrast, according to RNAcentral data consisting of 1D sequences (Sweeney et al., 2019), RNAs longer than 100 nucleotides account for 53.36% of all sequences. This indicates that while RNA sequences longer than 100 nucleotides are widely present, there is a significant scarcity of 3D structural data for them.

(3) In the RNAsolo dataset, among RNA structures with high resolution ($\leq 2.0$ Å), RNAs longer than 100 nucleotides account for only 3.60%; while at resolutions $> 2.0$ Å, RNAs longer than 100 nucleotides account for 24.62%. Table 7 shows detailed statistical information about the distribution.

All these findings indicate that experimental determination of large RNA structures presents significant challenges, leading to a much smaller number of large RNA structures than the small ones. Several papers published in top-tier journals have also highlighted this issue (Ma et al., 2022; Li et al., 2024a).

Given the difficulty in experimentally determining large RNA structures, we naturally consider using deep learning to predict RNA 3D structures. In this context, the evaluation task aims to select the candidate structure that most closely matches the experimentally determined native structure from the predicted RNA 3D structure ensembles. However, due to the limited number of available large RNA structures, it is not feasible to train a model on a large dataset composed of such RNAs to achieve strong evaluation performance. Therefore, we propose a dataset that leverages small RNA structures to train the evaluation model, then tests whether the evaluation capability can successfully generalize to larger RNA structures. This is precisely the *size generalization* problem we propose. We believe this problem is worthy of in-depth exploration.

Table 7: The distribution of RNAs of different sizes in the RNAsolo dataset.

| RNA Length Range | RNA $\leq 2.0$ Å | RNA $>2.0$ Å |
| --- | --- | --- |
| 0-100 | 610 | 648 |
| 100-200 | 11 | 118 |
| 200-300 | 0 | 19 |
| 300-400 | 0 | 53 |
| 400-500 | 0 | 15 |

## C    PROOF OF EQUIVARIANCE/INVARIANCE

In this section, we will prove that, the whole EquiRNA model is E(3)-invariant. And at first, we will prove that, such a conclusion can be drawn with three sub-conclusion as follow:

1. The modules $\texttt{EBN}(\cdot)$ and $\texttt{ERN}(\cdot)$ in are E(3)-equivariant;

2. The modules $\texttt{EB2R}(\cdot)$ and $\texttt{ER2B}(\cdot)$ are E(3)-equivariant;

3. The module $\texttt{ENN}(\cdot)$ is E(3)-equivariant;

**Theorem C.1.** *The whole EquiRNA model's E(3)-invariance can be drawn with the three sub-conclusion mentioned above.*

*Proof.* Consider a sequence composed of functions $\{\varphi_i : \mathcal{X}^{(i-1)} \to \mathcal{X}^{(i)}\}_{i=1}^N$ equivariant to a same group $G$, the equivariance lead to an interesting property that

$$\varphi_N \circ \cdots \circ \varphi_{i+1} \circ \rho_{\mathcal{X}^{(i)}}(g)\varphi_i \circ \cdots \circ \varphi_1 = \varphi_N \circ \cdots \circ \varphi_{j+1} \circ \rho_{\mathcal{X}^{(j)}}(g)\varphi_j \circ \cdots \circ \varphi_1,$$

holds for all $i, j = 1, 2, \ldots, N$ and $g \in G$, which means that the group elements $g$ can be freely exchanged in the composite sequence of equivariant functions. In particular, if one of the equivariant functions (*e.g.* $\varphi_k$) is replaced by an invariant function, the group element $g$ will be absorbed, that means

$$\varphi_N \circ \cdots \circ \varphi_k \circ \cdots \circ \varphi_{i+1} \circ \rho_{\mathcal{X}^{(i)}}(g)\varphi_i \circ \cdots \circ \varphi_1 = \varphi_N \circ \cdots \circ \varphi_1.$$

holds for all $g \in G$ but only $i = 1, 2, \ldots, k$. Although $\varphi_N \circ \cdots \circ \varphi_k$ is still equivariant, because the group elements must be input starting from $\varphi_1$, the overall $\varphi_N \circ \cdots \circ \varphi_1$ is still an invariant function.

Noted that the loss function in Eq. (4) only uses invariant feature $\boldsymbol{h}$, the whole model's invariance can be implied by the equivariance of all layers just as the three sub-conclusion.    □

Since $\texttt{EBN}(\cdot)$ and $\texttt{ERN}(\cdot)$ are all based on EGNN (Satorras et al., 2021), and the equivariance is obvious. We only need to prove the equivariance of Eqs. (2) and (3).

**Theorem C.2.** *The modules EB2R($\cdot$) and ER2B($\cdot$) are E(3)-equivariant.*

*Proof.* The key point is to prove the attention mechanism is E(3)-invariant. Since $\tilde{\boldsymbol{x}}_{r_i}^{l+1} = \vec{\boldsymbol{x}}_{r_i}^{l+1} - \vec{\boldsymbol{x}}_{r_c}^{l+1}$ and $\tilde{\boldsymbol{x}}_{b_i}^{l+1} = \vec{\boldsymbol{x}}_{b_i}^{l+1} - \vec{\boldsymbol{x}}_{b_c}^{l+1}$, all resluts based on them will be translation-invariance. And with the norm operator $\|\cdot\|$, we can build orthogonal-invariant features since $\|\boldsymbol{O}\vec{\boldsymbol{x}}\| = \|\vec{\boldsymbol{x}}\|$, and $\vec{\boldsymbol{x}}$ here can be replaced by $\tilde{\boldsymbol{x}}_{r_i}^{l+1}$ or $\tilde{\boldsymbol{x}}_{b_i}^{l+1}$. That means the attention mechanism is E(3)-invariant.

Other operators only linearly combine all Cartesian vectors without changing their own symmetries, so the module is E(3)-equivariant.    □

**Theorem C.3.** *The module ENN($\cdot$) is E(3)-equivariant.*

*Proof.* The key point is to prove the $\boldsymbol{m}_{ij}^n$ is E(3)-invariant. Since $W_{\boldsymbol{t}_i}^a \oplus W_{\tau_i}^n, W_{\boldsymbol{t}_j}^a \oplus W_{\tau_j}^n, \boldsymbol{P}_i, \boldsymbol{P}_j$ are all invariant embeddings from atom type ,nucleotide type and position (not coordinates), we only to show that the distance matrix $\boldsymbol{D}_{ij}$ is O(3)-invariant. The distance is O(3)-invariant can also be proved by unitarity of the norm, that means $\texttt{ENN}(\cdot)$ will not change the symmetry of the vector input. Since $\vec{\boldsymbol{x}}_{c_j}^{l+2}$ is the center of nucleotide $j$, $\vec{\boldsymbol{X}}_i^{l+2} - \vec{\boldsymbol{x}}_{c_j}^{l+2}$ outputs the relative position difference, which is translation-invariant and O(3)-equivariant, and that means the whole $\texttt{ENN}(\cdot)$ module is E(3)-equivariant.    □

## D    RRNASOLO DATASET

In this section, we provide details not covered in the main paper about our rRNAsolo dataset.

### D.1 GENERATION OF CANDIDATE STRUCTURES

Following the description of data collection, purification, and partitioning in the main paper, we first predict the secondary structure of all RNAs using (Zok et al., 2018). Then, with the primary and predicted secondary structures as inputs, we use the knowledge-based fragment assembly method FARFAR2 (Watkins et al., 2020) to generate candidate tertiary structures. Finally, we calculate the RMSD between these candidate structures and the native structure as the label for our task.

### D.2 THE CHOICE OF FARFAR2

Here, we chose to use FARFAR2 for dataset construction primarily for two reasons:

Firstly, the paper (Townshend et al., 2021) published in Science '21 uses FARFAR2 to construct their dataset, demonstrating that its generation quality has been practically verified and has undergone rigorous peer review, ensuring reliable quality.

Secondly, FARFAR2, as a classic RNA 3D structure prediction method, is widely recognized in the field. For instance, (Childs-Disney et al., 2022) published in Nature Reviews Drug Discovery '22 compares FARFAR2 to ROSETTA: "currently available programs, including FARFAR2 (RNA analogue of ROSETTA for protein prediction)" . The paper (Wong et al., 2023) published in Science '23 lists FARFAR2 alongside AlphaFold as representative sequence-to-structure models: "We anticipate that sequence-to-structure models, such as AlphaFold for proteins or FARFAR2 for RNAs."

Given FARFAR2's widespread recognition in the field and its use in constructing datasets in peer-reviewed papers, we decide to employ FARFAR2 to generate RNA candidates and build our rRNA-solo dataset. Additionally, other methods could also be used as long as they produce high-quality RNA candidate structures. In the future, we plan to further explore and optimize the rRNAsolo dataset to enhance the quality of candidate structures.

## E    CONFIGURATIONS

Table 8 presents the hyper-parameters of EquiRNA used in two experiments of this paper. Additionally, the results reported in our paper for other baselines are all obtained by retraining using publicly available code on rRNAsolo dataset. Each layer here consists of the Eq. (1), Eq. (2), and Eq. (3). Both our approach and all other baseline methods are trained and tested on a single NVIDIA A100-80G GPU.

Table 8: Hyper-parameters of EquiRNA. The nucleotide_template_size denotes the size of trainable template for each type of nucleotide, the atom_template_size denotes the size of trainable template for each type of atom, the hidden_size denotes the size of hidden states in our EquiRNA, and the n_layers denotes the number of layers in EquiRNA.

| Hyperparameter | rRNAsolo dataset | ARES dataset |
|---|---|---|
| Learning Rate | 1e-4 | 1e-4 |
| Epochs | 20 | 20 |
| nucleotide_template_size | 16 | 16 |
| atom_template_size | 16 | 16 |
| hidden_size | 128 | 128 |
| n_layers | 3 | 3 |
| K | 16 | 16 |
| M | 26 | 26 |

# F    MORE COMPREHENSIVE EXPERIMENTS AND ANALYSIS

## F.1    MORE EXPERIMENTS USING THE 1 BEST-SCORING MODEL AND 10 BEST-SCORING MODELS

To provide a more comprehensive analysis of our method compared to other methods, we present the results of all methods on the rRNAsolo and ARES datasets using the 1 best-scoring model and 10 best-scoring models in Table 9. As shown in the table, our method still achieves the best performance on the rRNAsolo dataset for the 10 best-scoring models, demonstrating the superior performance and robustness of our model. On the ARES dataset, our 10 best-scoring model is slightly inferior to the ARES method, but still achieves the second-best results, surpassing most of the other baseline methods. The 10 best-scoring setting evaluates the models' performance within the top 10 candidates, but it does not identify which candidate represents the best prediction. In contrast, the 1 best-scoring setting assumes that the top candidate is the best prediction, which better aligns with practical applications.

Table 9: Results of all methods on the rRNAsolo and ARES datasets using the 1 best-scoring model and 10 best-scoring models. Bold indicates the best method, while underline represents the second-best method.

|  | rRNAsolo | | | | ARES | | | |
|---|---|---|---|---|---|---|---|---|
|  | 1 best-scoring | | 10 best-scoring | | 1 best-scoring | | 10 best-scoring | |
|  | ER $\downarrow$ | R-ER $\downarrow$ | ER $\downarrow$ | R-ER $\downarrow$ | ER $\downarrow$ | R-ER $\downarrow$ | ER $\downarrow$ | R-ER $\downarrow$ |
| ARES | 22.57 | 0.84 | 16.19 | 0.32 | 12.33 | 1.02 | **8.42** | **0.43** |
| EGNN | 20.99 | 0.71 | 15.69 | 0.28 | 13.47 | 1.21 | 10.00 | 0.70 |
| PaxNet | 21.64 | 0.76 | 15.77 | 0.29 | 13.15 | 1.16 | 10.78 | 0.83 |
| dyMEAN | 20.91 | 0.71 | 15.95 | 0.30 | 13.56 | 1.23 | 9.35 | 0.59 |
| RDesign | 20.22 | 0.65 | 15.86 | 0.29 | 13.37 | 1.20 | 9.51 | 0.61 |
| EquiRNA | **18.22** | **0.48** | **15.30** | **0.25** | **11.74** | **0.93** | 8.99 | 0.52 |

## F.2    DETAILED ANALYSIS OF THE EXPERIMENTAL RESULTS

In this section, we endeavor to elucidate the potential factors contributing to the notable performance degradation of ARES on the rRNAsolo dataset.

(1) The strong performance of ARES reported in their paper might be due to the careful tuning of hyperparameters specific to their dataset. In the supplementary materials of the ARES paper, the authors stated, "We optimized several hyperparameters that specify characteristics of the training process: batch size, number of epochs, learning rate, and number of candidate structural models per RNA to feed in. We considered 100 sets of hyperparameter values. For each of these sets, we trained the network parameters." When applied to our rRNAsolo dataset using the same set of hyperparameters, ARES performs significantly worse, indicating reduced robustness across diverse scenarios. In contrast, our method, without extensive hyperparameter tuning and using the same hyperparameters across both datasets, consistently achieves superior performance.

(2) As mentioned in the final paragraph of § 2.2 and shown in Table 1, our rRNAsolo dataset features a broader range of RNA sizes, a larger number of RNAs, and more recently released data compared to the ARES dataset. Notably, the RNA size range overlaps between the training and test sets in the ARES dataset, whereas in our rRNA dataset, these ranges are distinct. Therefore, we believe our dataset presents a more challenging study for evaluating size generalization.

# G    REGARDING THE HIGH-LEVEL RMSD VALUES

Regarding the high-level RMSD values presented in the experimental tables, they are primarily due to the following two reasons:

Table 10: The results of rRNAsolo when considering molecular chirality.

| | Validation Set | | | | Test Set | | | |
|---|---|---|---|---|---|---|---|---|
| | ER ↓ | DR ↓ | R-ER ↓ | R-DR ↓ | ER ↓ | DR ↓ | R-ER ↓ | R-DR ↓ |
| EquiRNA with SBDD | 20.66 | 21.02 | 0.92 | 1.13 | 18.85 | 19.28 | 0.53 | 0.60 |
| EquiRNA | **19.77** | **19.62** | **0.84** | **0.99** | **18.22** | **17.79** | **0.48** | **0.47** |

Table 11: Ablation studies on rRNAsolo.

| | ER | DR | R-ED | R-DE |
|---|---|---|---|---|
| EGNN-like at Subunit-Level | 18.86 | 19.75 | 0.53 | 0.64 |
| Averaging at Nucleotide-Level | 19.87 | 18.66 | 0.62 | 0.55 |
| EquiRNA | **18.22** | **17.79** | **0.48** | **0.47** |

(1) The inherent difficulty of the task: As RNA size increases from small to large, the complexity and flexibility of RNA structures increase significantly, making the task more challenging.

(2) The quality of the RNA candidates: Since our task is RNA structure evaluation, our reported RMSD values largely depend on the candidate structures generated by FARFAR2 (Watkins et al., 2020) (FARFAR2 is a widely recognized method in the field, and for the rationale behind using FARFAR2, please refer to Appendix D.2). Here, we would like to further clarify the computation process of the RMSD metric for better understanding. For each RNA in the test set, we generate multiple candidate structures using FARFAR2. The task of the evaluation model is to score these candidate structures based on the model's predicted RMSD values, then select the one with the minimum predicted value, and finally report the ground-truth RMSD between the selected candidate structure and the native structure. Thus, the theoretical minimum of this value is **NOT** necessarily zero; instead, it is the lowest RMSD between the best candidate and the native structures. As our constructed dataset rRNAsolo is more challenging than the ARES dataset, the generated candidates by FARFAR2 (the same method used in the ARES paper to generate candidates) generally exhibit higher-level RMSD than those in the ARES dataset, leading to high-level RMSD reported in our experiments.

## H CONSIDERING MOLECULAR CHIRALITY

We attempt the approach from SBDD (Schneuing et al., 2022) by incorporating Eq. (6) and Eq. (7) of SBDD into our model to further account for molecular chirality. The results are shown in the Table 10. As can be seen, including chirality actually led to a slight performance drop. This is quite reasonable. In fact, our task is to predict the RMSD between candidate RNA structures and stable structures, which is an E(3)-invariant quantity and is insensitive to structural flipping. Therefore, for our task, an E(3)-invariant model is more appropriate than an SE(3)-invariant model. Of course, we agree that if the task involves predicting molecular properties sensitive to chirality, such as optical activity, then an SE(3)-invariant model would indeed be more suitable.

## I THE ABLATION STUDIES ON SUBUNIT-LEVEL AND NUCLEOTIDE-LEVEL

We conduct an experiment by replacing the attention mechanism with a simpler EGNN-like message-passing scheme in subunit-level. The results, presented in Table 11, show that the EGNN-like scheme performs worse compared to our original strategy. These findings highlight the effectiveness of the attention mechanism in capturing the intricacies of subunit-level interactions. Moreover, we conduct an experiment by averaging the atomic coordinates within each nucleotide. The experimental results, as shown in Table 11, indicate that this approach adversely affects the model's performance.

Table 12: The results of all four metrics on the first split.

| | Validation Set | | | | Test Set | | | |
|---|---|---|---|---|---|---|---|---|
| | ER ↓ | DR ↓ | R-ER ↓ | R-DR ↓ | ER ↓ | DR ↓ | R-ER ↓ | R-DR ↓ |
| RDesign | 13.04 | **9.56** | 0.92 | **0.46** | 16.36 | 16.12 | 0.86 | 1.44 |
| EquiRNA | **11.56** | 10.05 | **0.71** | 0.53 | **15.00** | **14.15** | **0.70** | **1.14** |

Table 13: The results of all four metrics on the second split.

| | Validation Set | | | | Test Set | | | |
|---|---|---|---|---|---|---|---|---|
| | ER ↓ | DR ↓ | R-ER ↓ | R-DR ↓ | ER ↓ | DR ↓ | R-ER ↓ | R-DR ↓ |
| RDesign | 15.86 | 13.84 | 1.22 | 1.78 | 13.01 | 11.16 | 1.07 | 0.83 |
| EquiRNA | **14.08** | **12.46** | **0.97** | **1.50** | **11.47** | **9.74** | **0.82** | **0.59** |

## J  MORE SPLITS ON RRNASOLO

We extend our study to a more general scenario where size generalization is not a constraint. Based on the clustering approach described in § 2.2, we randomly split the training, validation, and test sets at the cluster level. Due to time and resource limitations, we test three random split configurations, with each split containing 20k, 6k, and 6k samples in the training, validation, and test sets, respectively. The results, presented in Table 12, Table 13 and Table 14, demonstrate that our model consistently achieves superior performance across nearly all three splits, further underscoring its robustness and effectiveness. Furthermore, we also list the PDB IDs of the RNAs included in each split.

### J.1  FIRST SPLIT

Training: 4MGN_1_D, 2B57_1_A, 7EQJ_1_A, 7KVT_1_B, 3SKW_1_A, 4YB0_1_A, 5TPY_1_A, 7KVV_1_D, 6UC9_1_B, 7OAV_1_B, 2EET_1_A, 7OAX_1_B, 7OAX_1_D, 6TFF_1_A, 2GDI_1_Y, 6CC3_1_A, 3DS7_1_A, 3D2G_1_B, 4YB0_1_R, 6UET_1_A, 6UFH_1_B, 6Q57_1_A, 7KJU_1_A, 3OXE_1_B, 6PRV_1_C, 5C7U_1_B, 4FEL_1_B, 3OWI_1_A, 6QN3_1_A-B, 4FE5_1_B, 7D7W_1_A, 6UES_1_A, 6TFE_1_A, 7U4A_1_A, 5FK1_1_A, 3OWI_1_B, 7D82_1_A, 6UFJ_1_A, 3GER_1_A, 6DMC_1_B, 7OAV_1_D, 6WLQ_3_A, 2EEW_1_A, 4ENB_1_A, 3GAO_1_A, 2G9C_1_A, 4FEO_1_B, 2EES_1_A, 2QUW_1_B, 6TF3_1_A

Validation: 4PLX_1_A, 5OB3_1_A, 4JRC_1_A, 4K27_1_U, 2YIF_1_Z-X, 3R4F_1_A, 4WFM_1_B, 2YIE_1_X-Z, 6SVS_1_B, 7JRT_1_A, 4JRC_1_B, 6UFM_1_B, 3SLM_1_B, 6SVS_1_A, 7L0Z_1_G

Test: 5D5L_1_A, 7JRR_1_A, 6DN2_1_Y-X, 5V3I_1_A, 4JF2_1_A, 6DN1_1_Z-X, 7LYJ_1_A, 7K16_1_P, 6WTL_1_A, 7JJU_1_A-B, 6JQ5_1_B-A, 5D5L_1_C, 4P5J_1_A, 6WTR_1_A, 5D5L_1_B

### J.2  SECOND SPLIT

Training: 7OAV_1_B, 3SKR_1_A, 6DME_1_A, 3IVN_1_B, 3IVN_1_A, 2G9C_1_A, 4FE5_1_B, 4LVW_1_A, 4WFM_1_A, 7OAV_1_C, 6DN1_1_Z-X, 4WFM_1_B, 6UC9_1_B, 7D7Y_1_A, 3SKL_1_A, 4FEL_1_B, 6DMD_1_B, 4YB0_1_A, 5FKF_1_A, 1Y27_1_X, 6Q57_1_A, 3OWW_1_A, 3FO6_1_A, 4YB0_1_R, 6DN2_1_Y-X, 4TZX_1_X, 3SUX_1_X, 6TB7_1_A, 3OXE_1_B, 2YIF_1_Z-X, 6UC8_1_B, 2GDI_1_X, 4LVZ_1_A, 3SKL_1_B, 2XNZ_1_A, 6UFK_1_A, 4WFL_1_A, 3SKI_1_A, 6UFJ_1_C, 7D81_1_A, 3SKI_1_B, 3SKT_1_A, 4MGN_1_C, 7KVU_1_G, 6UFK_1_C, 5FK6_1_A, 3GOG_1_A, 6SVS_1_A, 2YIE_1_X-Z, 3OXD_1_A

Validation: 6N5N_1_A, 4JF2_1_A, 5TPY_1_A, 7JRR_1_A, 5D5L_1_A, 4PQV_1_A, 3FWO_1_B, 6CC3_1_A, 7K16_1_P, 7U4A_1_A, 7JJU_1_A-B, 5D5L_1_B, 2OIU_1_Q, 5D5L_1_D, 4R4V_1_A

Test: 7L0Z_1_G, 4EN5_1_A, 3R4F_1_A, 6V9D_1_B-E, 7LYJ_1_A, 4ENB_1_A, 4ENA_1_A, 4ENC_1_A, 5OB3_1_A, 4P5J_1_A, 1KXK_1_A, 5UNE_1_A-B, 4K27_1_U, 6JQ5_1_B-A, 3Q3Z_1_A

Table 14: The results of all four metrics on the third split.

| | Validation Set | | | | Test Set | | | |
|---|---|---|---|---|---|---|---|---|
| | ER ↓ | DR ↓ | R-ER ↓ | R-DR ↓ | ER ↓ | DR ↓ | R-ER ↓ | R-DR ↓ |
| RDesign | 15.61 | 15.41 | 1.05 | 1.14 | 16.68 | 15.16 | 1.03 | 1.24 |
| EquiRNA | **14.15** | **14.36** | **0.86** | **1.00** | **14.89** | **13.63** | **0.81** | **1.02** |

Table 15: The comparison with more leading models.

| | Validation Set | | | | Test Set | | | |
|---|---|---|---|---|---|---|---|---|
| | ER ↓ | DR ↓ | R-ER ↓ | R-DR ↓ | ER ↓ | DR ↓ | R-ER ↓ | R-DR ↓ |
| ProNet | 21.33 | 22.16 | 0.99 | 1.24 | 20.56 | 20.45 | 0.67 | 0.69 |
| Equiformer | 20.76 | 20.70 | 0.93 | 1.10 | 19.22 | 18.20 | 0.56 | 0.51 |
| EquiRNA | **19.77** | **19.62** | **0.84** | **0.99** | **18.22** | **17.79** | **0.48** | **0.47** |

## J.3 THIRD SPLIT

Training: 6WTR_1_A, 3DIR_1_A, 3IVN_1_B, 6WTL_1_A, 7D7Y_1_A, 4FEL_1_B, 6DMC_1_B, 7OAX_1_B, 4PLX_1_A, 3DIY_1_A, 4LX5_1_A, 3OXB_1_A, 7KVU_1_G, 7L0Z_1_G, 6UEY_1_C, 2EEW_1_A, 3OWW_1_A, 3GOT_1_A, 6N5N_1_A, 7EQJ_1_A, 6JQ5_1_B-A, 6TF3_1_A, 3OX0_1_A, 3SKW_1_B, 3SKW_1_A, 6N5K_1_A, 6UFG_1_B, 3OXE_1_A, 6N5P_1_A, 6N5T_1_A, 3SKT_1_A, 3SKL_1_B, 3SLQ_1_A, 3SKL_1_A, 6N5O_1_A, 7OAV_1_C, 6N5Q_1_A, 3SD3_1_A, 7KVT_1_B, 4TZX_1_X, 4FEN_1_B, 3SKR_1_A, 6UFM_1_A, 3D2V_1_B, 4FE5_1_B, 6UFJ_1_A, 4FEO_1_B, 6JQ6_1_U, 2CKY_1_A, 6N5S_1_A

Validation: 5FK1_1_A, 2OIU_1_P, 2QUW_1_D, 4K27_1_U, 2QUW_1_B, 3Q3Z_1_V, 6DN1_1_Z-X, 5FKF_1_A, 3Q3Z_1_A, 2QUS_1_A, 5FJC_1_A, 6V9D_1_B-E, 2QUS_1_B, 4P8Z_1_A, 1KXK_1_A

Test: 6DN2_1_Y-X, 3SLM_1_B, 3VRS_1_A, 4WFL_1_A, 4ENB_1_A, 4YB0_1_A, 4YB0_1_R, 7JJU_1_A-B, 4ENC_1_A, 4PQV_1_A, 4WFM_1_A, 4EN5_1_A, 4ENA_1_A, 4R4V_1_A, 6WLQ_3_A

## K DISCUSSION ON THE RECENT SCALABLE EQUIVARIANCE METHODS

SE(3)-Hyena (Moskalev et al., 2024), VN-EGNN (Sestak et al., 2024), and Ponita (Bekkers et al., 2023) employ global context (the SE(3)-Hyena operator, virtual nodes, and global group actions, respectively) to enhance the scalability of equivariant neural networks. These approaches represent excellent strategies for further improving general-purpose models. In contrast, our EquiRNA is a specialized design tailored to leverage the prior knowledge of RNA nucleotide structures, enabling the effective application of equivariant neural networks to large RNA molecules. This specialization allows EquiRNA to address the unique challenges of modeling RNA, distinguishing it from the more general approaches discussed.

## L THE COMPARISON WITH MORE LEADING MODELS

There are various strong baselines like SphereNet (Liu et al., 2022), Equiformer (Liao & Smidt, 2022), MACE (Batatia et al., 2022), and ProNet (Wang et al., 2022). We conduct experiments with Equiformer and ProNet. The results are shown in Table 15. Equiformer demonstrates strong performance, likely due to its incorporation of higher-order information. ProNet, however, does not perform as well. We suspect that the original ProNet paper is for proteins and it computes Euler angles using alpha carbon, carbon, and nitrogen atoms, which are not strictly corresponding to the atoms in RNA. In our reproduction, we substituted these with P, C4', and C3' atoms, which might have influenced ProNet's performance to some extent. Overall, our EquiRNA still achieves the best performance, substantiating our claims.

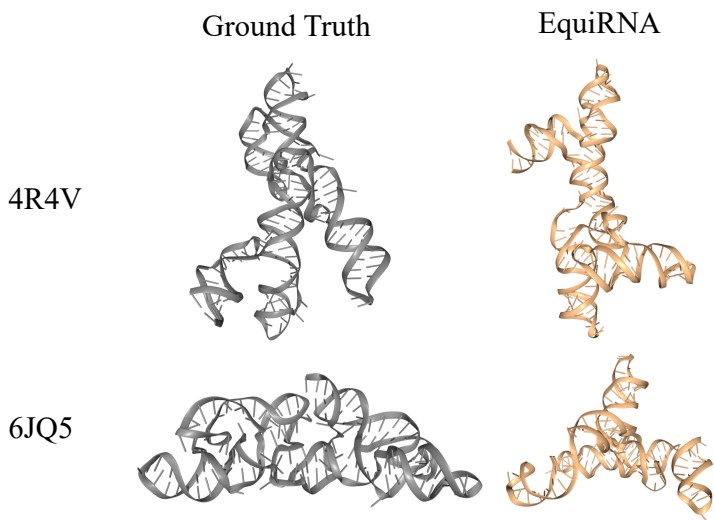

Figure 6: The visualization of bad cases.

Table 16: The results of rRNAsolo when considering noise.

| | Validation Set | | | | Test Set | | | |
| --- | --- | --- | --- | --- | --- | --- | --- | --- |
| | ER ↓ | DR ↓ | R-ER ↓ | R-DR ↓ | ER ↓ | DR ↓ | R-ER ↓ | R-DR ↓ |
| EquiRNA w noise | 20.44 | 20.68 | 0.90 | 1.09 | 19.51 | 18.94 | 0.59 | 0.57 |
| EquiRNA | **19.77** | **19.62** | **0.84** | **0.99** | **18.22** | **17.79** | **0.48** | **0.47** |

## M    THE ANALYSIS OF SOME ERROR CASES

We have presented two RNA samples in Fig. 6 where EquiRNA performs poorly. We observe that both samples exhibit significant structural symmetry, which we hypothesize may contribute to the suboptimal performance. Symmetry could result in a phenomenon akin to "systematic extinction" in crystallography (Janner & Janssen, 1977), where information in certain directions is weakened or lost. Although RNA structures are not perfectly symmetric, this might affect the model's ability to effectively learn directional information. Indeed, prior literature (Joshi et al., 2023; Lawrence et al., 2024) suggests that equivariant neural networks may encounter challenges when dealing with symmetric structures.

## N    CONSIDERING NOISE

Since noise is common in experimental structure determination/sequencing (Xu et al., 2024a), we conduct the experiments to evaluate our model when considering noise. We apply Gaussian noise $\mathcal{N}(\mathbf{0}, 0.01 \cdot \mathbb{E}(\|\vec{x} - \vec{x}_c\|) \cdot \mathbf{I})$ to the coordinates of all training RNAs, where $x$ represents the 3D coordinate of each atom in RNA, and $x_c$ represents the mean coordinate of all the atoms in one RNA. The results on rRNAsolo are shown in the Table 16. It is observed that adding noise leads to a decline in model performance. Excitingly, our EquiRNA with noise is still better than other methods that are trained from clean data.

## O    THE FIGURE FOR THE GRAPH STRUCTURE

We illustrate the graph structures in § 2.1 more clearly with the Fig. 7.

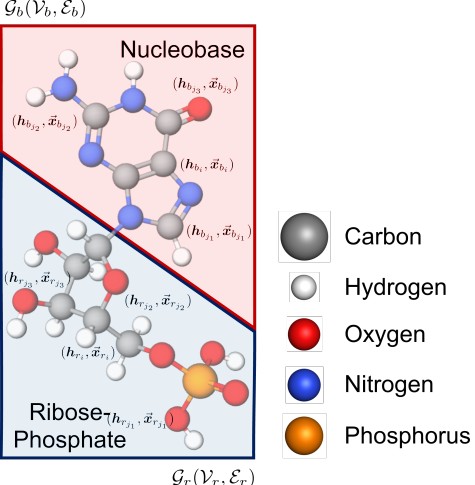

Figure 7: RNA distribution of RNAsolo.

## P THE DETAILED EXPLANATION OF THE IMPLEMENTATION OF OTHER BASELINES

ARES and PaxNet were originally designed for RNA structure evaluation. We use their official code for our experiments.

As for EGNN, the original paper models small molecules at the atomic level. When adapting it to our task, we model RNA at the full-atom level and directly use their official code.

For dyMEAN, the original paper models proteins using amino acids as basic units. When adapting it to our task, we replaced amino acids with nucleotides and modeled RNA using nucleotides as basic units, employing their official code.

For RDesign, the original paper is developed for RNA inverse folding tasks and considers only backbone atoms. Since RNA representations in the original method primarily rely on backbone dihedral angles and distance information, we retain the original settings by using backbone atoms to represent RNA and employ the invariant feature h calculated from the method for the final RMSD prediction. We use their official code for our experiments.

For GET, the original paper models proteins hierarchically by capturing intra-amino acid and inter-amino acid interactions via blocks. When adapting it to our task, we replace amino acids with nucleotides and use their official code.

## Q FINETUNED RESULTS ON A NEW DATASET GENERATED BY RHOFOLD.

In this section, we attempt to utilize the recently published RhoFold (Shen et al., 2024) method as a new candidate structure generator to construct a new dataset and finetune the models on this dataset.

Predicting 3D structures with RhoFold requires both 1D fasta sequences and MSA data. However, the RNAsolo dataset used in our experiments does not provide the corresponding MSA data. To overcome this, we first search for similar sequences for each RNA using RNAcentral (https://rnacentral.org/) and BLAST (https://blast.ncbi.nlm.nih.gov/Blast.cgi). We then perform multiple sequence alignment with Infernal (Nawrocki & Eddy, 2013) and MAFFT (Katoh et al., 2002) to generate the required MSA data files. Following the methodology outlined in the original paper (Shen et al., 2024), we randomly sample from the MSA data to ensure the uniqueness of each generated candidate structure. Finally, we compute the RMSD values between the candidate structures and their native counterparts using the pair_fit function in PyMOL, which is based on the Kabsch algorithm (Kabsch, 1976).

We randomly select 20 RNAs from the training set in our rRNAsolo and generate 400 candidate structures for each RNA using RhoFold, following the process described above. This results in a dataset containing 8,000 samples. We then finetune both RDesign and our EquiRNA model on this new dataset. The experimental results are presented in Table 17.

Table 17: The finetuned results on the new dataset generated by RhoFold.

| | Validation Set | | | | Test Set | | | |
|---|---|---|---|---|---|---|---|---|
| | ER ↓ | DR ↓ | R-ER ↓ | R-DR ↓ | ER ↓ | DR ↓ | R-ER ↓ | R-DR ↓ |
| RDegisn | 21.99 | 20.79 | 1.05 | 1.11 | 20.22 | 19.34 | 0.65 | 0.60 |
| RDegisn+finetune | 20.86 | 21.02 | 0.94 | 1.31 | 21.53 | 20.45 | 0.75 | 0.69 |
| EquiRNA | 19.77 | 19.62 | 0.84 | 0.99 | **18.22** | **17.79** | **0.48** | **0.47** |
| EquiRNA+finetune | **18.87** | **18.54** | **0.76** | **0.88** | 18.78 | 18.25 | 0.53 | 0.51 |

The results show that after finetuning, our EquiRNA model achieves improved performance of all four metrics on the validation set, while RDesign shows better performance only in terms of the ER and R-ER metric. However, EquiRNA showed a slight decline in test set performance, possibly due to bias introduced when using only FARFAR2 to generate candidate structures. Since the structures generated by RhoFold follow a different distribution from those generated by FARFAR2, the model's performance on the test set could be partially influenced by this bias. Notably, EquiRNA exhibits a much smaller performance drop compared to RDesign on the test set (*e.g.*, 0.56 vs. 1.5 in terms of ER), suggesting that our model is more robust across different candidate structure generators.

## R  BROADER IMPACTS

Our in-depth investigation into more accurate methods for modeling and evaluating the 3D structures of RNA positively impacts both our understanding of the fundamental mechanisms of biological activities and the discovery of RNA-targeted drugs.

## S  LIMITATIONS

Relative to the abundance of data for proteins, extant RNA datasets are substantially inadequate to train highly accurate prediction and scoring models, especially for large RNAs where the intricate structural variations often lead to higher RMSD values. It is our aspiration to delve deeper into the prediction and scoring tasks of large RNAs in our subsequent research endeavors.

