# OpenReview forum: "Size-Generalizable RNA Structure Evaluation by Exploring Hierarchical Geometries"
_ICLR.cc/2025/Conference — ICLR 2025 Poster_

### Official Review · Reviewer_sFcQ · 2024-10-30

**Soundness:** 4
**Presentation:** 3
**Contribution:** 3
**Rating:** 6
**Confidence:** 4

**Summary:**

The paper introduces a three-level hierarchical equivariant graph neural network (EquiRNA) designed to efficiently capture the geometries of RNA, enabling it to generalize effectively across a wide range of RNA sizes. EquiRNA demonstrates promising results in evaluating RNA 3D structures.

**Strengths:**

1.	The proposed hierarchical equivariant graph for RNA structures is novel.
2.	The results on the size generalization dataset are compelling.

**Weaknesses:**

1.	The assessment is not informative (see Q1).
2.	The baselines compared with EquiRNA are used across various tasks, but their adaptation for structure evaluation remains unclear (see Q2).
3.	While EquiRNA performs well on datasets of RNAs with unseen sizes, its performance on the datasets splitting irrespective of size remains unclear.

**Questions:**

1.	The paper presents comparisons with several models, but there are other models, such as SphereNet[1], Equiformer[2], MACE[3], and ProNet[4], that show better performance. A more extensive comparison with existing methods would enhance the paper. Although the experimental results highlight the proposed method's effectiveness, a more thorough comparison with other leading models could further substantiate the paper's claims.
2.	A detailed explanation of the implementation of several baselines is needed. For example, RDesign was originally developed for RNA inverse folding tasks and only includes backbone atoms. How was RDesign adapted for RNA structure evaluation in this paper?
3.	The author emphasizes the importance of the size-generalization task but should include an experiment that evaluates dataset splitting independent of size.

[1] Liu, Yi, et al. "Spherical message passing for 3d molecular graphs." International Conference on Learning Representations (ICLR). 2022.
[2] Liao, Yi-Lun, and Tess Smidt. "Equiformer: Equivariant graph attention transformer for 3d atomistic graphs." arXiv preprint arXiv:2206.11990 (2022).
[3] Batatia, Ilyes, et al. "MACE: Higher order equivariant message passing neural networks for fast and accurate force fields." Advances in Neural Information Processing Systems 35 (2022): 11423-11436.
[4] Wang, Limei, et al. "Learning hierarchical protein representations via complete 3d graph networks." arXiv preprint arXiv:2207.12600 (2022).

---

> ### Author Response · Authors · 2024-11-22
> **Responses to Reviewer sFcQ**
>
> We are grateful for your comprehensive feedback! Below we provide point-by-point responses and outline the corresponding changes made to improve our work.
>
> > **Q1:The paper presents comparisons with several models, but there are other models, such as SphereNet[1], Equiformer[2], MACE[3], and ProNet[4], that show better performance. A more extensive comparison with existing methods would enhance the paper. Although the experimental results highlight the proposed method's effectiveness, a more thorough comparison with other leading models could further substantiate the paper's claims.**
>
> **A1**: Thank you for your suggestion! As you mentioned, comparing with more leading models would further validate our claims. Due to time constraints, we conduct experiments with two recent methods, Equiformer and ProNet. To further complete our paper, we have cited MACE and SphereNet in the revised paper. The results are shown in Table A. Equiformer demonstrates strong performance, likely due to its incorporation of higher-order information. ProNet, however, does not perform as well. We suspect that the original ProNet paper is for proteins and it computes Euler angles using alpha carbon, carbon, and nitrogen atoms, which are not strictly correspondint to the atoms in RNA. In our reproduction, we substituted these with P, C4', and C3' atoms, which might have influenced ProNet's performance to some extent. Overall, our EquiRNA still achieves the best performance, substantiating our claims.
>
>  **Table A: The comparison with more leading models.**
>
> |            | Validation Set |   |        |        |  Test Set     |       |        |        |
> | ---------- | -------------- | -------- | ------ | ------ | ----- | ----- | ------ | ------ |
> |            | ER ↓           | DR  ↓    | R-ER ↓ | R-DR ↓ | ER ↓  | DR  ↓ | R-ER ↓ | R-DR ↓ |
> | ProNet     | 21.33          | 22.16    | 0.99   | 1.24   | 20.56 | 20.45 | 0.67   | 0.69   |
> | Equiformer | 20.76          | 20.70    | 0.93   | 1.10   | 19.22 | 18.20 | 0.56   | 0.51   |
> | EquiRNA    | **19.77**          | **19.62**    | **0.84**   | **0.99**   | **18.22** | **17.79** | **0.48**   | **0.47**   |
>
> > **Q2: A detailed explanation of the implementation of several baselines is needed. For example, RDesign was originally developed for RNA inverse folding tasks and only includes backbone atoms. How was RDesign adapted for RNA structure evaluation in this paper?**
>
> **A2**: Thank you for your suggestion! We provide detailed descriptions of the implementation for each baseline method below and have added this content to Appendix P.
>
> - ARES and PaxNet were originally designed for RNA structure evaluation. We use their official code for our experiments.
>
> - As for EGNN, the original paper models small molecules at the atomic level. When adapting it to our task, we model RNA at the full-atom level and directly use their official code.
>
> - For dyMEAN, the original paper models proteins using amino acids as basic units. When adapting it to our task, we replaced amino acids with nucleotides and modeled RNA using nucleotides as basic units, employing their official code.
>
> - For RDesign, the original paper is developed for RNA inverse folding tasks and considers only backbone atoms. Since RNA representations in the original method primarily rely on backbone dihedral angles and distance information, we retain the original settings by using backbone atoms to represent RNA and employ the invariant feature h calculated from the method for the final RMSD prediction. We use their official code for our experiments.
>
> - For GET, the original paper models proteins hierarchically by capturing intra-amino acid and inter-amino acid interactions via blocks. When adapting it to our task, we replace amino acids with nucleotides and use their official code.
>
> > **Q3: The author emphasizes the importance of the size-generalization task but should include an experiment that evaluates dataset splitting independent of size.**
>
> **A3**: Thank you for your suggestion! As you mentioned, although our study focuses on size generalization tasks, conducting an experiment independent of size would further strengthen our conclusions. Based on the clustering approach described in Appendix C.1, we randomly split the training, validation, and test sets at the cluster level. Due to time and resource limitations, we tested three random split configurations, with each split containing 20k, 6k, and 6k samples in the training, validation, and test sets, respectively.  The results, presented in Appendix J of the revised version, demonstrate that our model consistently achieves superior performance across nearly all three splits, further underscoring its robustness and effectiveness. For your reference, Appendix J also lists the PDB IDs of the RNAs included in each split.

---

> > ### Comment · Reviewer_sFcQ · 2024-11-27
> > **I am  confused about random split?**
> >
> > According to my experience, data split is very crucial for RNA task evaluation, how did you split at  the cluster level？ why random split is still used? the authors should evaluate the data split with considering both the size and strucutre.

---

> > > ### Author Response · Authors · 2024-11-27
> > >
> > > Thank you for your response! We apologize for not clearly describing the random split in our previous responses to Q3. Below, we provide a detailed explanation of the random split process. We adopt the clustering approach mentioned in USAlign [1], sorting RNAs by the number of nucleotides in descending order. Starting with the largest RNA as the cluster representative, the TM-score is computed with the rest using the USAlign algorithm. If the score is higher than 0.45, it's categorized into the same cluster as the largest RNA; otherwise, the largest RNA forms its own cluster. Repeat the process with the largest RNA of the remaining RNAs until all RNAs are clustered. After such clustering, the TM-Score between any two clusters is less than 0.45, indicating that they have different global structural topologies. To ensure that the training, validation, and test sets all contain small RNAs (50-100nt) and large RNAs (100-200nt), we perform data split for small and large RNAs separately.
> > >
> > > First, we identify clusters containing RNA sizes between 50nt and 100nt and perform N rounds of selection. In each round, we randomly select one cluster and randomly pick X RNAs from it (where X ranges from 1 to 5) to avoid selecting too many RNAs from a single cluster, which could compromise the diversity of RNA structures. Importantly, at the beginning of each round, we calculate the remaining number of RNAs needed and, in the final round, select only the exact number required. We then remove remaining RNAs in these selected clusters to prevent data leakage. Similarly, we randomly select RNA from the remaining clusters which contains 50nt-100nt RNAs to form the validation and test sets, and delete all other RNAs in these clusters afterward.
> > >
> > > Next, we perform the same process with the remaining clusters containing RNA sizes between 100nt and 200nt. We randomly select clusters, pick RNAs of 100nt-200nt for the training set, and remove these clusters to prevent data leakage. The same method is applied to construct the validation and test sets using 100-200nt clusters.
> > >
> > > Notably, because RNA selection is based on clusters, the training, validation, and test sets do not share similar global structural topology, effectively preventing data leakage.
> > >
> > > We hope this explanation provides you with a better understanding of our dataset splitting process!
> > >
> > > Zhang C et al. US-align: universal structure alignments of proteins, nucleic acids, and macromolecular complexes, Nature methods 2022.

---

> > > ### Author Response · Authors · 2024-12-01
> > > **Looking forward to your responses**
> > >
> > > Dear Reviewer sFcQ,
> > >
> > > As the rebuttal deadline is approaching, we would like to kindly ask if our responses have addressed your concerns. We deeply value your expert insights, which are essential for enhancing our work, and we eagerly anticipate your constructive feedback and guidance.
> > >
> > > Best regards,
> > >
> > > The Authors

---

### Official Review · Reviewer_egW8 · 2024-10-31

**Soundness:** 3
**Presentation:** 2
**Contribution:** 3
**Rating:** 8
**Confidence:** 4

**Summary:**

The paper presents EquiRNA, a hierarchical Geometric Graph Neural Network model designed for size-generalizable RNA structure evaluation. Addressing challenges in capturing full RNA geometries and generalizing across different RNA sizes, EquiRNA uses a three-level architecture—atom, subunit, and nucleotide—to model RNA’s structural complexity effectively. It introduces a size-insensitive K-nearest neighbor sampling strategy to enhance robustness when generalizing from small to large RNAs, and a new dataset, rRNAsolo, for thorough evaluation. Experimental results show EquiRNA outperforms SOTA methods, providing a robust new baseline for RNA 3D structure prediction and evaluation.

**Strengths:**

1.	Hierarchical Architecture for Size Generalization: By modeling RNA structure at atom, subunit, and nucleotide levels, EquiRNA effectively captures detailed RNA geometries and adapts well to RNAs of varying sizes, addressing the common challenge of generalizing from small to large RNAs.

2.	Innovative Sampling Strategy: The size-insensitive K-nearest neighbor sampling enhances robustness by exposing the model to diverse structural variations, allowing better generalization to larger RNAs not seen during training.

3.	Comprehensive New Dataset: The rRNAsolo dataset, covering a broader range of RNA sizes and types, provides a more rigorous benchmark for size generalization, demonstrating EquiRNA’s superior performance over state-of-the-art methods in real-world scenarios.

**Weaknesses:**

See Questions.

**Questions:**

1.	Better to have a figure when explaining the graph structrure in 3.1
2.	In line 53, what’s the definition of “resolution”? Is it the threshold of deciding if there is an edge between two atoms?
How and why to select “resolution higher than 2 A”?
3.	Why don’t use average for $\overrightarrow{\boldsymbol{X}}^{l+2} \in \mathbb{R}^{3 \times\left(N_r+N_b\right)}$, instead of a concatenation?

4.	For atom-level and subunit-level layers, do they apply only on 4 types of nucleotides (AUGC), or each nucleotide respectively?

5.	Line 878 shows the number of layers for both datasets are 3. If I stack Eq. 1, Eq.2, and Eq.3, are they together 1 layer or 3 layers?
6.	Equation 4 is confusing, and I’m not clear with the overall task.

(1)	The task is to predict RMSD between candidate and native structures. So on RNAsolo’s website (https://rnasolo.cs.put.poznan.pl), the provided structures are ground-truth (native) structures, right? Then what’s the candidate structures and how did you get them? Based on my understanding, there are multiple 3D strucure prediction tools for one RNA sequence, then what are the prediction tools? And, how many candidate structures are compared to the ground-truth structure?

(2)	In Eq.4, is $\hat{\boldsymbol{h}}_i^{l+3}$ the ground truth label for each nucleotide? How did you get that? I understand the ground truth structure is given by a PDB file, but how was it converted into the ground truth for each nuclotide?

(3) Why is $\hat{h}_i$ ground truth? Isn't it the feature? If it is ground truth here, then the notation is duplicated.

(4) Why to use features h to predict final RMSD? Why not X? What’s the function of row 3 in Eq.3 if X is not used?

I'd appreciate it if the authors can describe the overall task, candidate structures, ground truth labels comprehensively.

7. Eq. 1 and Eq.2 have the similar form with EGNN. The difference is just on different levels (atom level, sub-unit level). Is there any intuition why to use RBF?

8.	Code is not available.

---

> ### Author Response · Authors · 2024-11-22
> **Responses to Reviewer egW8 (Part 1/2)**
>
> Thank you for your thorough and insightful review! Please find below our detailed responses and the associated changes we have implemented in the manuscript.
>
> > **Q1: Better to have a figure when explaining the graph structrure in 3.1**
>
> **A1:** Thank you for your suggestions! We have now illustrated the graph structures in Section 3.1 more clearly with a figure in Appendix O.
>
> > **Q2: In line 53, what’s the definition of “resolution”? Is it the threshold of deciding if there is an edge between two atoms? How and why to select “resolution higher than 2 A”?**
>
> **A2:** Sorry for the insufficient clarity. Here, "resolution" refers to a key metric for evaluating the quality of structural data obtained using experimental methods such as cryo-EM, X-ray crystallography, or NMR spectroscopy. The definition of 'resolution' is the minimum distance at which two adjacent atoms can be distinguished.
>
> It reflects the capability of experimental methods to distinguish molecular details. Typically, a value smaller  than 2 Å is considered high resolution, allowing clear identification of most atomic positions and making it suitable for structural biology studies [1,2].
>
> > **Q3: Why don’t use average for $\vec{\boldsymbol{X}}^{l+2}\in\mathbb{R}^{3\times(N\_r+N\_b)}$, instead of a concatenation?**
>
> **A3:** Thanks. In RNA molecules, the atoms in the Ribose-phosphate and nucleobase significantly influence RNA structure and properties. Using concatenation maximizes the retention of this information, allowing subsequent nucleotide-level message passing to fully utilize these atom-level features. We also replace concatenation with averaging and conduct ablation experiments. The results, shown in Table A, indicate that this substitution impairs model performance.
>
> Table A: Ablation studies on rRNAsolo.
>
> |                               | ER ↓  | DR  ↓ | R-ER | R-DR |
> | ----------------------------- | ----- | ----- | ---- | ---- |
> | Averaging at Nucleotide-Level | 19.87 | 18.66 | 0.62 | 0.55 |
> | EquiRNA                       | **18.22** | **17.79** | **0.48**| **0.47** |
>
> > **Q4: For atom-level and subunit-level layers, do they apply only on 4 types of nucleotides (AUGC), or each nucleotide respectively?**
>
> **A4:** Yes, for atom-level and subunit-level layers, they apply on each nucleotide respectively.
>
> > **Q5: Line 878 shows the number of layers for both datasets are 3. If I stack Eq.1, Eq.2, and Eq.3, are they together 1 layer or 3 layers?**
>
> **A5:** Sorry for the insufficient clarity. Eq.1, Eq.2, and Eq.3 collectively form only 1 layer. We have corrected this potentially misleading description in the paper.
>
> > **Q6.1: The task is to predict RMSD between candidate and native structures. So on RNAsolo’s website (https://rnasolo.cs.put.poznan.pl), the provided structures are ground-truth (native) structures, right? Then what’s the candidate structures and how did you get them? Based on my understanding, there are multiple 3D strucure prediction tools for one RNA sequence, then what are the prediction tools? And, how many candidate structures are compared to the ground-truth structure?**
>
> **A6.1**: Yes, the structures provided on the RNAsolo website are all ground-truth structures. Candidate structures refer to the set of RNA 3D structures predicted based on 1D and 2D inputs. We used FARFAR2 [3] to generate the candidate structures based on the 1D and 2D structures provided in RNAsolo. FARFAR2 generates 3D candidate structures via fragment assembly. For each RNA in the train/val/test sets, we generated 400 candidate structures. Details of the generation process and the reasons for selecting FARFAR2 can be found in Appendix C.
>
> > **Q6.2: In Eq.4, is $\hat{h}^{l+3}_i$ the ground truth label for each nucleotide? How did you get that? I understand the ground truth structure is given by a PDB file, but how was it converted into the ground truth for each nuclotide?**
>
> **A6.2**: Thank you for your valuable feedback! The original description in our paper contains inaccuracies, and we sincerely apologize for the oversight. When calculating the loss, we do not use the ground-truth label of each nucleotide. Instead, we sum up the features $h^{l+3}\_{i}$ of all nucleotides within the RNA, pass the result through an MLP to obtain the predicted RMSD value, and finally compute the loss with the RNA's corresponding ground-truth label. We have revised this in the paper accordingly. Thank you again for your careful observation and contributions to improving our paper!

---

> ### Author Response · Authors · 2024-11-22
> **Responses to Reviewer egW8 (Part 2/2)**
>
> > **Q6.3:  Why is $\hat{h}_i$ ground truth? Isn't it the feature? If it is ground truth here, then the notation is duplicated.**
>
> **A6.3**: Thank you for this valuable feedback! Our original description was unclear, and using $\hat{h}_i$ could indeed cause misunderstanding. We have replaced it with $y$ and made corresponding revisions in the main text.
>
> > **Q6.4: Why to use features h to predict final RMSD? Why not X? What’s the function of row 3 in Eq.3 if X is not used?**
>
> **A6.4**: The label we need to predict is the RMSD value between each candidate structure and the native structure, which is an E(3)-invariant variable. Hence, we choose to use the E(3)-invariant feature $H$ instead of the E(3)-equivariant coordinate $X$, to predict the final RMSD.
>
> Although the final $X$ is not directly related to the predicted RMSD, during the updates in Eq. (3), the atomic coordinates of each nucleotide in $X$ are iteratively updated to compute the matrix $D$. This is then used to calculate edge features $m\_{ij}$ and other related features, which are subsequently used to update both $X$ and $H$.
>
> > **Q7: Eq. 1 and Eq.2 have the similar form with EGNN. The difference is just on different levels (atom level, sub-unit level). Is there any intuition why to use RBF?**
>
> **A7**: Eq. 1 is indeed similar, but in Eq. 2, we introduce an attention mechanism to learn adaptive weighting coefficients ($α\_{ij}$), capturing the complex relationships in RNA and their varying importance. RBF has been widely used in several studies, such as DimeNet [4] and GET [5]. We consider RBF as a form of distance discretization that models various types of interactions between atoms, thereby capturing more comprehensive information.
>
> > **Q8: Code is not available.**
>
> **A8**: We have published the pipeline used for dataset construction, inference code, and test datasets via an anonymous link: https://anonymous.4open.science/r/EquiRNA-9D02. The complete code and dataset will be made publicly available after the paper is accepted.
>
> [1] Nakane T, Kotecha A, Sente A, et al. Single-particle cryo-EM at atomic resolution[J]. Nature 2020.
>
> [2] Dubach V R A, Guskov A. The resolution in X-ray crystallography and single-particle cryogenic electron microscopy[J]. Crystals 2020.
>
> [3] Watkins A M et al. FARFAR2: Improved De Novo Rosetta Prediction of Complex Global RNA Folds. Structure 2020.
>
> [4] Gasteiger J et al. Directional message passing for molecular graphs. ICLR 2020.
>
> [5] Kong X et al. Generalist equivariant transformer towards 3d molecular interaction learning. ICML 2024.

---

> > ### Comment · Reviewer_egW8 · 2024-11-22
> >
> > Thanks for your detailed responses.
> >
> > RA 4:
> > I’m still unclear about this question. Let me make it clearer. Imagine we have a 10-nucleotide “AUGCAUGCAU”. Do you have 4 different atom-level and subunit-level layers for each type of nucleotide, or do you have 10 different atom-level and subunit-level layers for each nucleotide?
> > In line 94 “We propose EquiRNA, a hierarchical equivariant graph neural network, tailored to address the size generalization challenge by reusing the representations of nucleotides.” Based on my understanding, you only have 4 different atom-level and subunit-level layers for each type of nucleotide. Then how do you differentiate the same type of nucleotide appearing in different positions in a long RNA sequence? For example,  “A” appears in the first 5th, and 9th positions in the sequence “AUGCAUGCAU”.
> >
> >
> > RA 6.1:
> > 1)	You mentioned, “We used FARFAR2 [3] to generate the candidate structures based on the 1D and 2D structures provided in RNAsolo.”  Could you point out to me where can I find the 2D structures provided by RNAsolo?  Are these 2D structures also experimentally validated?
> > 2)	“For each RNA in the train/val/test sets, we generated 400 candidate structures.” So all 400 candidate structures are generated by FARFAR2? Why can this tool generate 400 candidates? Do you use different parameters to generate different candidate structures?
> > 3)	Some more recent 3D prediction methods [1, 2, 3] only need 1D sequence as the input, and the deep learning methods are proven to have better performances [2]. Why don’t you choose to use more recent and promising methods? I raise this question because if all 400 candidate filters are generated by FARFAR2, training on your rRNAsolo dataset can lead to bias on the structures generated by FARFAR2. All structures generated by other tools may lie out of the distribution.
> >
> > RA 6.2:
> > A 6.2 didn’t answer my question. I understand how you get the
> > $MLP(\sum_{i=1}^{N_{\mathrm{nu}}} \boldsymbol{h}_i^{l+3})$.
> >
> > If I assume $\hat{y} = MLP\left(\sum_{i=1}^{N_{\mathrm{nu}}} \boldsymbol{h}_i^{l+3}\right)$, is  $\hat{y}$ here a RMSD value, which is a scalar?
> >
> > And what is $y$ here? Is it a PDB file, or vector, or scalar? And how did you convert the PDB file to something else?
> >
> > References:
> > [1] NuFold: a novel tertiary RNA structure prediction method using deep learning with flexible nucleobase center representation.
> > [2] State-of-the-RNArt: benchmarking current methods for RNA 3D structure prediction.
> > [3] Integrating end-to-end learning with deep geometrical potentials for ab initio RNA structure prediction.

---

> ### Author Response · Authors · 2024-11-23
> **Responses to Reviewer egW8**
>
> Thank you very much for your feedback. We apologize for the unclear and misleading points caused in our previous responses. We provide more explanations below.
>
> > **RA 4:**
>
> Sorry for misunderstanding your question in Q4 in our previous responses. We initially thought you were asking whether each nucleotide conducts the atom-level and subunit-level layers.  Now, we understand that you were actually asking if different nucleotides apply different atom-level and subunit-level layers. In our model, all nucleotides—regardless of whether they are of the same type—share the same atom-level and subunit-level layers. Even so, we can still differentiate the same type of nucleotide appearing in different sequential positions (such as "A" at the 5th and 9th positions of your example 'AUGCAUGCAU', ). This is because the input to our model includes the 3D positions of atoms in each nucleotide, enabling the model to differentiate nucleotides based on their unique 3D structural representations. We hope this clarification helps address your concerns.
>
> > **RA 6.1:**
>
> 1) The RNAsolo dataset does not directly provide the 2D structure data for RNA. Our initial response may not have been sufficiently rigorous. Specifically, we use the 1D data provided in the RNAsolo dataset and apply the existing 2D structure prediction tool RNAPDBee [1] to generate 2D structures.
>
> 2) Yes, all 400 candidate structures are generated using FARFAR2 [2]. During the FARFAR2 execution, we specify the number of candidate structures to generate by setting the parameter "nstruct=400" in the configuration file. In particular, FARFAR2 "implements a special set of Monte Carlo moves for nucleotides in stacked Watson-Crick pairs" [2] to generate multiple candidate structures. The complete pipeline is provided in the anonymous link https://anonymous.4open.science/r/EquiRNA-9D02, where you can find the generation commands in the `generate.sh` file and the parameter settings in the `flags_1.txt` file.
>
> 3) Thank you for your suggestion! We provided a detailed explanation of our choice to use FARFAR2 for generating candidate structures in Appendix C.3, summarized as follows: 1. Multiple studies [3,4] have highlighted that FARFAR2 holds a position in RNA prediction tasks comparable to ROSETTA or AlphaFold in protein prediction, demonstrating its high level of recognition. 2. Previous works [5] have successfully used FARFAR2 for RNA structure prediction and candidate structure generation, validating its effectiveness. Based on these reasons, we chose FARFAR2 to generate our candidate structures.
> At the same time, we agree with your observation that recent deep learning methods have shown promising results in generating 3D structures.
> As you mentioned, relying solely on structures generated by FARFAR2 may introduce bias during training. **During the rebuttal period, we will make every effort to include experimental results involving the candidate structures generated using DRfold [6] as you suggested**. However, as this requires some time, we kindly ask for your understanding if we are unable to provide these results in time.
>
> > **RA 6.2:**
>
> Sorry for confusing you in our previous replies. Here, $\hat{y}$ is the predicted RMSD value by our model and a scalar.  The symbol $y$ is the ground-truth RMSD value between the candidate structure and the native structure, which is also a scalar, calculated by FARFAR2. We will highlight the above points in the revised version for better clarity.
>
> We hope these responses address your concerns. Please feel free to let us know if anything remains unclear.
>
> [1] Rnapdbee 2.0: multifunctional tool for rna structure annotation, Nucleic acids research 2018.
>
> [2] FARFAR2: Improved De Novo Rosetta Prediction of Complex Global RNA Folds, Structure 2020.
>
> [3] Targeting RNA structures with small molecules, Nature Reviews Drug Discovery 2022.
>
> [4]  Leveraging artificial intelligence in the fight against infectious diseases, Science 2023.
>
> [5] Geometric deep learning of RNA structure, Science 2021.
>
> [6] Integrating end-to-end learning with deep geometrical potentials for ab initio RNA structure prediction, Nature Communications 2023.

---

> ### Author Response · Authors · 2024-12-01
> **New experimental results**
>
> Dear Reviewer egW8,
>
> As promised in our previous response, we are pleased to share the new experimental results, which include the use of an alternative candidate structure generator in addition to FARFAR2.
>
> Initially, we attempted to use DRfold [1] as suggested. However, we found that generating a single candidate structure using the official DRfold code took several hours, making it impractical to generate a sufficient number of candidate structures within the time constraint. As a result, we explored other recent deep learning based methods and identified the newly published RhoFold [2] as a suitable alternative. The detailed generation process is outlined below.
>
> Predicting 3D structures with RhoFold requires both 1D fasta sequences and MSA data. However, the RNAsolo dataset [3] used in our experiments does not provide the corresponding MSA data. To overcome this, we first search for similar sequences for each RNA using RNAcentral [4] and BLAST [5]. We then perform multiple sequence alignment with Infernal [6] and MAFFT [7] to generate the required MSA data files. Following the methodology outlined in the original paper [2], we randomly sample from the MSA data to ensure the uniqueness of each generated candidate structure. Finally, we compute the RMSD values between the candidate structures and their native counterparts using the pair_fit function in PyMOL, which is based on the Kabsch algorithm [8].
>
> We randomly select 20 RNAs from the training set and generate 400 candidate structures for each RNA using RhoFold, following the process described above. This results in a dataset containing 8,000 samples. We then finetune both RDesign and our EquiRNA model on this new dataset. The experimental results are presented in Table B below.
>
> **Table B: The finetuned results on the new dataset generated by RhoFold.**
> |       |        |     |      |      |      |     |    |     |
> | :--------------- | :------------- | :----- | :----- | :----- | :------- | :----- | :----- | :----- |
> |    |      |       |      |     |        |     |     |     |
> |    | Validation Set |      |      |     | Test Set |     |      |      |
> |      | ER ↓           | DR ↓   | R-ER ↓ | R-DR ↓ | ER ↓     | DR ↓   | R-ER ↓ | R-DR ↓ |
> | RDesign          | 21\.99         | 20\.79 | 1\.05  | 1\.11  | 20\.22   | 19\.34 | 0\.65  | 0\.60  |
> | RDesign+finetune | 20\.86         | 21\.02 | 0\.94  | 1\.31  | 21\.53   | 20\.45 | 0\.75  | 0\.69  |
> | EquiRNA          | 19\.77         | 19\.62 | 0\.84  | 0\.99  | **18\.22**   | **17\.79** | **0\.48**  | **0\.47**  |
> | EquiRNA+finetune | **18\.87**         | **18\.54** | **0\.76**  | **0\.88**  | 18\.78   | 18\.25 | 0\.53  | 0\.51  |
> |                 |                |        |        |        |          |        |        |        |
>
> The results show that after finetuning, our EquiRNA model achieves improved performance of all four metrics on the validation set, while RDesign shows better performance only in terms of the ER and R-ER metric. However, performance on the test set declines slightly for EquiRNA, which may be linked to your earlier remark that "candidate structures generated entirely by FARFAR2 might introduce bias." Since the structures generated by RhoFold follow a different distribution from those generated by FARFAR2, the model’s performance on the test set could be partially influenced by this bias. **Notably, EquiRNA exhibits a much smaller performance drop compared to RDesign on the test set (e.g., 0.56 vs. 1.5 in terms of ER), suggesting that our model is more robust across different candidate structure generators.**
>
> Once again, thank you for your insightful suggestions, which have greatly improved our paper! We are in the process of uploading the new dataset generated by RhoFold, and you can access it via the anonymous link (https://anonymous.4open.science/r/EquiRNA-9D02). We will definitely include the new results in the revised paper and further enhance the manuscript based on your suggestions. We hope this response adequately addresses your concerns.
>
>
> [1] Li Y et al. Integrating end-to-end learning with deep geometrical potentials for ab initio RNA structure prediction. Nature Communications, 2023.
>
> [2] Shen T et al. Accurate RNA 3D structure prediction using a language model-based deep learning approach. Nature Methods, 2024.
>
> [3] Adamczyk B et al. RNAsolo: a repository of cleaned PDB-derived RNA 3D structures. Bioinformatics, 2022.
>
> [4] https://rnacentral.org/
>
> [5] https://blast.ncbi.nlm.nih.gov/Blast.cgi
>
> [6] Nawrocki E P et al. Infernal 1.1: 100-fold faster RNA homology searches. Bioinformatics, 2013.
>
> [7] Katoh K et al. MAFFT: a novel method for rapid multiple sequence alignment based on fast Fourier transform. Nucleic acids research, 2002.
>
> [8] Kabsch W. A solution for the best rotation to relate two sets of vectors. Acta Crystallographica Section A: Crystal Physics, Diffraction, Theoretical and General Crystallography, 1976.
>
> Best regards,
>
> The Authors

---

> ### Comment · Reviewer_egW8 · 2024-12-03
>
> Thank you for your response. Most of my concerns have been addressed. I hope the authors can incorporate the explanations into the paper to make it clearer, especially for readers who are not familiar with this area. Additionally, the new experimental results should be included. Therefore, I am raising my score to 8 in support of the paper’s acceptance.

---

> > ### Author Response · Authors · 2024-12-03
> >
> > We are glad that most of your concerns have been resolved! Following your suggestions, we will incorporate the relevant explanations and the new experimental results into our paper. Thank you again for your valuable feedback and effort during the review process!

---

### Official Review · Reviewer_Pevo · 2024-11-01

**Soundness:** 2
**Presentation:** 2
**Contribution:** 2
**Rating:** 6
**Confidence:** 4

**Summary:**

The paper introduces EquiRNA, a model which has three-level hierarchical architecture—atom-level, subunit-level, and nucleotide-level—inspired by the biological composition of RNAs. The model is designed to efficiently model RNA structures while handling RNAs of varying sizes, referred to as the size generalization problem in the paper. The paper introduces a new dataset curated from public database and benchmarks the proposed model against this dataset and an existing ARES dataset outperforming existing methods for the task of predicting RMSD between candidate and native structures, used for scoring and ranking all candidates to identify the one most
closely matching the native structure.

**Strengths:**

1. The paper proposes three-level E(3)-equivariant message-passing design for modeling RNA 3D structures that captures atom, subunit, and nucleotide interactions effectively. This is inspired by the biology of RNA and sensible contribution.
2. The paper showcases an application of EGNN architecture to model the interactions within different levels of RNA. Although it is a straightfoward application of EGNN, I find it addresses a meaningful modeling aspect respecting the prior knowledge from biology. Size-insensitive K-nearest neighbor sampling is also intended to boost the model’s performance on varied RNA sizes, addressing a common challenge in RNA modeling.
3. Introducing rRNAsolo benchmark is useful for future development in this area.
5. The paper is well-written for most parts, with ideas explained in a structured manner.

**Weaknesses:**

Although the paper tackles an important problem, there exist several avenues for improvement.
1. Given that the paper is meant to address the size generalization aspect of RNA structure evaluation, the experiments are carried out mostly on relatively smaller RNA sequences (~200 nucleotides). In most practical applications, the RNA sequences can easily range from hundreds to even thousands of nucleotides. To strengthen the claim that the proposed model can generalize to larger size RNAs, is it possible to showcase performance on larger RNA sizes or to discuss any technical limitations that prevented testing on very large RNAs?

2. Though the K-nearest neighbor sampling helps, there might still be overfitting to smaller RNAs, as training primarily relies on these samples. It is possible that I may have misunderstood but I am not sure how K-nearest neighbor sampling strategy guarantees the size generalization of the model. Given that this is claimed as the core contribution of the paper, it is remarkably brief and covered within 1 small paragraph. A detailed explanation around this will help improve readability.

3. The design, being based on EGNN, might face issues with large RNAs common in drug discovery due to increasing complexity in both memory and computation. Famously, SE(3)-Transformer (Fuchs et al., 2020), Equiformer (Liao et al., 2022) has been utilized for modeling large molecules such as proteins in models such as AlphaFold, there also have been recent works such as SE(3)-Hyena (Moskalev et al. 2024) using long-convolutional models for equivariant modeling of large-scale systems. Such architectures may directly model all-atom resolution thus providing an alternative to hierarchical modeling at backbone level as described in the paper. Potentially, the authors may consider an ablation study comparing their hierarchical approach to an all-atom model for the largest RNAs in their dataset or may provide a complexity analysis comparing their approach to SE(3)-Transformer, Equiformer and SE(3)-Hyena for large RNA modeling to justify architectural choice.

4. The paper does not seem to explore cases where EquiRNA performs poorly or where errors arise, missing an opportunity to analyze model limitations and provide insights into potential biases or weaknesses in real-world RNA structures. Can the authors provide specific examples where EquiRNA performs poorly compared to other methods or any patterns in the types of RNA structures or features that lead to higher errors?

5. The use of learned atom and nucleotide templates is innovative, but I wonder if this can also hamper generalizability. I imagine that generalizability is partly tied to specific template choices, which might not always capture structural diversity in RNAs. If new RNA structures or variations appear, the model may not capture these well unless it’s retrained or the templates are updated. If the model is reliant on pre-learned templates, it may struggle with RNAs that exhibit novel conformations or atypical atom interactions.

6. Experimental mehods like cryo-EM, X-ray crystallography, or NMR spectroscopy—for obtaining RNA structures—can produce data with resolution limits. This has mentioned in the introduction but I seem to have missed how the model performance changes when applied to   more/less noisy data (in terms of structural resolution).

**Questions:**

I have posed my questions/concerns in the weaknesses section already. Here is a summary and few suggestions which may help address some of my concerns.
1. Can the authors show how does the model handle variations in RNA structures due to experimental noise or biological variability (performance with different structural resolutions)? Also since noise is common in experimental structure determination/sequencing, can we consider testing the model's performance on RNA structures with simulated noise or structural variability to assess its robustness?
2. How sensitive is the model’s performance to the choice of templates? Is there a risk that template selection introduces bias or limits the generalizability of the model?
3. Has the model been evaluated on a variety of RNA structures, particularly those with unusual or rare structural motifs, to ensure generalizability?
4. Is it possible to run experiments showcasing the model's ability to truly generalize on large RNA sequences common in drug discovery (many hundreds to thousands of nucleotides), such as those derived from high-throughput sequencing or large RNA viruses?
5. I am not sure how K-nearest neighbor sampling strategy guarantees the size generalization of the model. Can this section be expanded upon?
6. Is it possible to showcase some error cases where the model fails and if that reveals some insights?
7. For the model to be usable and further developments to happen in this space, releasing code and curated datasets will be useful. Can the authors release them?

---

> ### Author Response · Authors · 2024-11-22
> **Responses to Reviewer Pevo (Part 1/3)**
>
> We deeply appreciate your valuable feedback! Detailed responses to your questions are provided below, along with the corresponding revisions incorporated into our paper.
>
> > **W1 && Q4: Is it possible to run experiments showcasing the model's ability to truly generalize on large RNA sequences common in drug discovery (many hundreds to thousands of nucleotides), such as those derived from high-throughput sequencing or large RNA viruses?**
>
> **A1:** Thank you for your suggestion! Our dataset already surpasses existing datasets such as ARES [1] in scale (200 nts vs. 150 nts). However, it is currently challenging to expand it to include more and larger RNAs. The primary limitation lies in the efficiency of candidate structure generation. Using FARFAR2 [2], generating 400 candidate structures for a ~200-nt RNA on a single-core CPU requires one to two weeks, and the computational time increases significantly as the RNA length grows. We are actively exploring more efficient and accurate methods for candidate structure generation, but FARFAR2 remains our first choice as the gold standard in the field. Details of our choice of FARFAR2 can be found in Appendix C.3. Additionally, once candidate structures are generated, modeling, training, and inference times also increase substantially with larger RNA structures. We have prioritized efficient modeling for larger RNAs as a future research direction. As you mentioned, this aligns more closely with real-world scenarios and holds great practical significance.
>
> > **W2 && Q5: I am not sure how K-nearest neighbor sampling strategy guarantees the size generalization of the model. Can this section be expanded upon?**
>
> **A2:** Thank you for your comment! The K-nearest neighbor sampling strategy is inspired by the concept of dropout. During training, for each nucleotide, we randomly and equally sample K neighbors from M-nearest neighbors to model their relationships. This approach trains the model with fewer neighbors (K) but effectively captures a broader spatial information (M). During testing, the model generalizes by leveraging the information learned from M neighbors. In general, as the number of nucleotides in RNA increases, more neighbor relationships are needed for edge construction, making generalization more challenging. By randomly masking edges during training, we aim to enable the model to learn features independent of nucleotide count, enhancing robustness to RNA size variations and improving generalization to larger RNAs. This is similar to the approach in [3], where randomly sampling subgraphs during training improves the model's robustness to size variations.
>
> > **W3: Potentially, the authors may consider an ablation study comparing their hierarchical approach to an all-atom model for the largest RNAs in their dataset or may provide a complexity analysis comparing their approach to SE(3)-Transformer, Equiformer and SE(3)-Hyena for large RNA modeling to justify architectural choice.**
>
> **A3:** Thank you for your insightful comments! First, we would like to emphasize that although our model employs a hierarchical architecture, it is fully capable of modeling at all-atom resolution, since atomic information is preserved across all three levels of message passing (as already stated in Line 082), ensuring detailed and accurate representation. Additionally, the complexity of models based on EGNN is actually lower than that of SE(3)-Transformer and Equiformer. The latter rely on high-degree tensors, which significantly increase computational demands.
>
> As per your suggestion, we reproduced Equiformer and reported the results in Table A. To address its high memory cost, we limited the number of neighbors to 30 for Equiformer during the experiments. The results indicate that Equiformer generally outperforms other baselines, likely due to its all-atom modeling approach and incorporation of higher-degree information. However, its performance remains notably worse than that of our EquiRNA. For instance, on the largest RNA in the test set (6GYV-1-A), the predicted RMSD values for EquiRNA and Equiformer are 17.14 and 22.82, respectively. By modeling hierarchical geometric patterns at full-atom resolution, our method demonstrates a superior ability to represent RNA structures compared to Equiformer, which was originally designed for general molecular modeling. This highlights the effectiveness of our specialized approach for RNA-specific tasks.
>
> Table A: The comparison with Equiformer on rRNAsolo.
>
> |            | Validation Set |       |      |        | Test Set      |       |        |        |
> | ---------- | -------------- | -------- | ------ | ------ | ----- | ----- | ------ | ------ |
> |            | ER ↓    | DR  ↓    | R-ER ↓ | R-DR ↓ | ER ↓  | DR  ↓ | R-ER ↓ | R-DR ↓ |
> | Equiformer | 20.76 | 20.70    | 0.93   | 1.10   | 19.22 | 18.20 | 0.56   | 0.51   |
> | EquiRNA    | **19.77**   | **19.62**    | **0.84**   | **0.99**   | **18.22** | **17.79** | **0.48**   | **0.47**   |

---

> ### Author Response · Authors · 2024-11-22
> **Responses to Reviewer Pevo (Part 2/3)**
>
> We have included an analysis of the complexities of SE(3)-Transformer, Equiformer, and SE(3)-Hyena in Appendix G and Table B (see below). In this analysis,  $N\_{a}, K\_{a}, C, L$represent the number of atoms in each RNA, the number of nearest atom neighbors, the channel size of high-degree tensors, and the maximum degree of tensors, respectively. To facilitate comparison, we also present the complexity of our EquiRNA, where $N\_{nu}, K\_{nu}, n\_a, d\_a, d\_{nu}$  denote the number of nucleotides in each RNA, the number of nearest nucleotide neighbors, the number of atoms per nucleotide, the embedding size of the atom template, and the embedding size of the nucleotide template, respectively.
>
> In our experiments, we usually have $(n^2\_a+n\_a(d\_a+d\_{nu}))N\_{nu}K\_{nu}<N_{a}K_{a}$, which demonstrates the lower computational complexity of EquiRNA compared to other methods. This holds even for approaches involving high-degree tensors (i.e., $L>1$), further showcasing the efficiency of our method.
>
> Table B: Complexity Analysis.
>
> | Methods           | Complexity                                            |
> | ----------------- | ----------------------------------------------------- |
> | ARES              | $O(N\_{a}K\_{a}C^{3}L^{6})$                           |
> | SE(3)-Transformer | $O(N\_{a}K\_{a}C^{3}L^{6})$                           |
> | Equiformer        | $O(N\_{a}K\_{a}C^{3}L^{6})$                           |
> | SE(3)-Hyena       | $O(N\_{a}(K\_{a}+logN\_{a})C^{3}L^{6})$                |
> | EquiRNA           | $O(n\_{a}N\_{a}+(n^2\_a+n\_a(d\_a+d\_{nu}))N\_{nu}K\_{nu})$ |
>
> > **W4 && Q4: Is it possible to showcase some error cases where the model fails and if that reveals some insights?**
>
> **A4:** Thank you for your nice advice! We have present two RNA samples in Appendix M where EquiRNA performs poorly. We observe that both samples exhibit significant structural symmetry (globally or locally), which we hypothesize may contribute to the suboptimal performance. Symmetry could result in a phenomenon akin to "systematic extinction" in crystallography [4], where information in certain directions is weakened or lost. Although RNA structures are not perfectly symmetric, this might affect the model's ability to effectively learn directional information. Indeed, prior literature [5,6] suggests that equivariant neural networks may encounter challenges when dealing with symmetric structures.
>
> > **W5 && Q2: The use of learned atom and nucleotide templates is innovative, but I wonder if this can also hamper generalizability. How sensitive is the model’s performance to the choice of templates? Is there a risk that template selection introduces bias or limits the generalizability of the model?**
>
> **A5:** Thank you for your insightful comment! Our templates are designed to be general and have been crafted to encompass a wide range of typical RNA cases. They account for four atom types (C, O, N, P) and four nucleotide types (A, C, G, U).
>
> In particular, for the nucleotide template ($W\_k^n$), we have enhanced generalizability by assigning different templates to the same atom type in different nucleotides (Lines 315-317). This design enables our model to capture the unique attributes of each atom within varying nucleotide contexts, effectively reflecting the nuanced characteristics of RNA structures. Additionally, increasing the number of channels (i.e., $d\_a$ and $d\_n$) could further enhance the model's ability to represent more diverse structures.
>
> The ablation studies presented in Table 4 demonstrate that incorporating both the atom template ($W\_k^a$) and nucleotide template ($W\_k^n$) significantly improves performance. Considering that we address the size generalization problem—where test structures differ markedly from training structures—this performance boost highlights the strong generalizability of our designed templates.
>
> > **W6: Experimental mehods like cryo-EM, X-ray crystallography, or NMR spectroscopy—for obtaining RNA structures—can produce data with resolution limits. This has mentioned in the introduction but I seem to have missed how the model performance changes when applied to more/less noisy data (in terms of structural resolution).**
>
> **A6:** Thank you for your detailed observation! As mentioned in the introduction of our paper (Figure 2), we highlight that small RNAs are typically resolved at high resolution (< 2 Å), while large RNAs tend to be resolved at lower resolution (> 2 Å). This distinction underscores the current limitations of experimental methods in determining the 3D structures of large RNAs. Motivated by this, our work aims to address the size generalization problem, leveraging high-quality small RNA structures to improve the estimation of large RNA structures. Nevertheless, as you mentioned, it is valuable to check the model performance changes when applied to more/less noisy data. We will reply to you in Q1 below.

---

> ### Author Response · Authors · 2024-11-22
> **Responses to Reviewer Pevo (Part 3/3)**
>
> > **Q1: Can the authors show how does the model handles variations in RNA structures due to experimental noise or biological variability (performance with different structural resolutions)? Can we consider testing the model's performance on RNA structures with simulated noise or structural variability to assess its robustness?**
>
> **Answers to Q1:** Thank you for this great advice! In response, we apply Guassian noise $\mathcal{N}(\boldsymbol{0}, 0.01\cdot \mathbb{E}(\\|\vec{\boldsymbol{x}}-\vec{\boldsymbol{x}}\_c\\|)\cdot \boldsymbol{I})$ to the coordinates of all training RNAs, where $\vec{\boldsymbol{x}}$ represents the 3D coordinate of each atom in RNA, and $\vec{\boldsymbol{x}}\_c$ represents the mean coordinate of all the atoms in one RNA. The results on rRNAsolo are shown in the table below. It is observed that adding noise leads to a decline in model performance. Excitingly, our EquiRNA with noise is still better than other methods that are trained from clean data.
>
> Table C: The results of rRNAsolo when considering noise.
>
> |                 | Validation Set |       |        |        | Test Set      |       |        |        |
> | --------------- | -------------- | -------- | ------ | ------ | ----- | ----- | ------ | ------ |
> |                 | ER ↓           | DR  ↓    | R-ER ↓ | R-DR ↓ | ER ↓  | DR  ↓ | R-ER ↓ | R-DR ↓ |
> | EquiRNA w noise | 20.44          | 20.68    | 0.90   | 1.09   | 19.51 | 18.94 | 0.59   | 0.57   |
> | EquiRNA         | **19.77**          | **19.62**    | **0.84**   | **0.99**   | **18.22** | **17.79** | **0.48**   | **0.47**   |
>
> > **Q3: Has the model been evaluated on a variety of RNA structures, particularly those with unusual or rare structural motifs, to ensure generalizability?**
>
> **Answers to Q3:** Given the rigorous construction process of our dataset (as described in Lines 202-204), we believe that our model has been extensively trained, validated, and tested on diverse sets of RNA structures. The RNAs were initially clustered based on structural similarity, ensuring that different clusters represent a variety of patterns. Subsequently, we enumerated all clusters for dataset partitioning, which guarantees comprehensive coverage of this diversity. In this way, the generalizability of our method is well-supported. Further details can be found in Section 3.2 of the main text and Appendix C.
>
> > **Q7: For the model to be usable and further developments to happen in this space, releasing code and curated datasets will be useful. Can the authors release them?**
>
> **Answers to Q7:** Thank you for your suggestion! We have published the pipeline used for dataset construction, inference code, and test datasets via an anonymous link: https://anonymous.4open.science/r/EquiRNA-9D02. The complete code and dataset will be made publicly available after the paper is accepted.
>
>
>
> [1] Townshend R J L et al. Geometric deep learning of RNA structure. Science 2021.
>
> [2] Watkins A M et al. FARFAR2: Improved De Novo Rosetta Prediction of Complex Global RNA Folds. Structure 2020.
>
> [3]Buffelli D et al. Sizeshiftreg: a regularization method for improving size-generalization in graph neural networks.
>
> [4] Janner A, Janssen T. Symmetry of periodically distorted crystals, Physical Review B.
>
> [5] Joshi C K, Bodnar C, Mathis S V, et al. On the expressive power of geometric graph neural networks, ICML23.
>
> [6] Lawrence H, Portilheiro V, Zhang Y, et al. Improving equivariant networks with probabilistic symmetry breaking, ICML24 Workshop on Geometry-grounded Representation Learning and Generative Modeling.

---

> ### Comment · Reviewer_Pevo · 2024-11-22
> **Major concerns remain unaddressed and more concerns arise based on responses**
>
> Thank you for your responses but my major concerns remain unaddressed and the authors seem to simply avoid some of the comments from me and other reviewers.
> 1) When the paper claims size generalization to arbitrary sized RNAs and shows experiments on  extremely small RNAs. Claiming that candidate structure generation as an issue seems like avoiding the experiments. Also I fail to understand the claim of FARFAR2 as SOTA for candidate structure generation given that it was published in 2020. There have been many more algorithms proposed for the same since (see DeepFoldRN 2022; RoseTTAFoldNA 2024;  trRosettaRNA 2023 and many others) which are also faster than FARFAR2. Showing results on ~200 nucleotide RNAs and claiming that their method is size generalizable seems exaggeration.
>
> Similarly, for many other questions, any details are missing about how and why some of the observations reported hold true (some examples below).
>
> 2) How does the proposed model handle noise? The response says that the proposed method is stil better than other methods but they only show comparison with their own model with and without noise. How does their model handle noise to begin with when it was not specifically optimized for it?
>
> 3) The authors complelety avoided my question about rare structural motifs by simply saying that they believe their model is size-genralizable. This was not my question and as mentioned above, based on the experiments, I contest the claim about size-generalization in the first place (see point 1).
>
> 4) Similarly, for comparing their architecture against EquiFormer, the authors limit the number of neighbors to 30 which beats the purpose of all-atom modeling and show that their architecture is better. Why not use architectures such as FastEGNN?
>
> 5) Regarding my question on template selection introducing bias, the authors in their response have simply repeated what they have in the paper originally. I see other reviewers also questioning this selection.
>
> 6) Response to other reveiwers about adapting RDesign: The authors respond saying that they use backbone atoms to represent RNA in RDesign. There are various ways to extract or represent the RNA backbone in molecular modeling and structural biology without clear consensus (see Richardson et al., 2008, RNA backbone: Consensus all-angle conformers and modular string nomenclature). Have the authors considered the choice of backbone representation in modeling and performance of their model and baselines?
>
> Overall, reading the response and reviews from other reviewers, I believe this paper has many limitations and may benefit from more comprehensive experiments, better justification of architectural choices and thorough comparison with other baselines and architectures. In the current form, the paper does not pass the bar for acceptance at ICLR.
>
> ***Based on the unsatisfactory responses to my questions and similar questions raised by other reviewers about the method, I have several concerns about the paper and I have decided to lower my score.***

---

> > ### Author Response · Authors · 2024-11-24
> > **Responses to Reviewer Pevo (Part 1/2)**
> >
> > Thank you very much for your response. We apologize for not adequately addressing your concerns in our previous replies. We have been making every effort to respond to your questions and have no intention of avoiding your concerns. We are happy to provide further clarification and hope for your understanding.
> >
> > > **Q1 & Q3**
> >
> > You may have misunderstood our claims. We are not attempting to solve the problem of size generalization for arbitrary RNA sizes, as this is a highly challenging task. As an initial exploration, our work focuses on training the model on smaller RNAs (50-100 nts) and testing its generalization ability on larger RNAs (100-200 nts).
> >
> > Regarding why we chose FARFAR2 for candidate structure generation, we have provided detailed explanations in Appendix C.3, which can be summarized as follows: 1. Multiple studies [1,2] have highlighted that FARFAR2 holds a position in RNA prediction tasks comparable to ROSETTA or AlphaFold in protein prediction, demonstrating its high level of recognition. 2. Previous works [3] have successfully used FARFAR2 for RNA structure prediction and candidate structure generation, validating its effectiveness. Based on these reasons, we chose FARFAR2 to generate our candidate structures.
> >
> > At the same time, we agree with your observation that there have been many more algorithms proposed for the RNA 3D structure prediction. However, DeepFoldRN is still a preprint and has not been officially published. RoseTTAFoldNA primarily focuses on predicting protein–nucleic acid complexes. The open-source code page for trRosettaRNA on the Nature website is currently inaccessible. Nevertheless, during the rebuttal period, we will make every effort to include experimental results involving candidate structures generated using the published and open-source method DRfold [4]. However, as this requires some time, we kindly ask for your understanding if we are unable to provide these results in time.
> >
> > > **Q2**
> >
> > We apologize for the lack of clarity in our previous response to Q1. What we intended to convey is that, in Table C from our earlier response, our method EquiRNA achieves better results on noisy training data compared to other methods listed in Table 1 of the paper. And these methods in Table 1 were all trained on non-noisy RNA data.
> >
> > Additionally, our model cannot handle cases where noise exists only in the test structures, as this was not considered during the design phase. However, this is not the main focus of our paper. Similarly, most studies in this field do not explicitly address noise when designing model architectures. But we also notice that prior work [5] has studied the topic of noise in this area, and we have discussed and cited it in our paper. We appreciate the reviewer for pointing this out and will also consider it as a potential direction for future research.
> >
> >
> > > **Q3**
> >
> > We have made every effort to address your questions and had no intention of avoiding any concerns. We apologize if our previous response caused any misunderstandings. As an initial exploration, we acknowledge that our method may face challenges in generalizing to rare structural motifs. However, this limitation is common to all learning-based approaches when there is a significant discrepancy between training and testing distributions. Our original response intended to suggest that if our method demonstrates generalization ability on the size generalization problem, it likely captures shared information about nucleotides as fundamental units across both short and long RNAs, indicating some level of generalization. If rare structural motifs are still composed of the same four types of nucleotides, our method may also exhibit generalization potential.
> >
> > Furthermore, could you clarify which rare structural motifs you are referring to? We would appreciate it if you could provide some examples. Thank you!
> >
> > > **Q4**
> >
> > In the original EquiFormer paper, the experiments were mainly conducted on datasets of small molecules, including QM9, MD17, and OC20. However, the number of atoms in RNAs is much larger. Directly adopting the full-atom setting causes OOM issues during training. What we want to emphasize is that EquiFormer employs higher-order tensors, which result in significant computational and memory overhead. As noted by the EquiFormer authors in their GitHub issue "I guess reducing $L\_{\max}$ from 2 to 1 would hurt the most in most of cases", removing higher-order tensors may have the most detrimental impact on model performance. Therefore, we retain higher-order tensors while reducing the number of neighbors to prevent OOM issues.
> >
> > > **Why not use architectures such as FastEGNN?**
> >
> > Since FastEGNN fully connects virtual nodes, assigning a virtual node to each of the $n$ nucleotides in an RNA would result in $n^{2}$ complexity, significantly increasing computational costs. Moreover, its CoM virtual node initialization may not be suitable for this problem that clearly has a sequence prior.

---

> > > ### Author Response · Authors · 2024-11-24
> > > **Responses to Reviewer Pevo (Part 2/2)**
> > >
> > > > **Q5**
> > >
> > > We apologize if our previous response did not adequately address your concerns. In our model, the template design is indeed specific; however, these templates are learnable and can be considered as part of the model parameters. We acknowledge that template-based methods may face challenges in generalization, but this is a common issue for all learning-based approaches.
> > >
> > > > **Q6**
> > >
> > > Thank you for your suggestion. As you noted, there are indeed diverse approaches to modeling the RNA backbone. In our approach, we use information from all heavy atoms in the RNA to model the overall structure, aiming to capture the most comprehensive information possible. For fairness, all other baselines in our experiments adopt the same setting, except for RDesign. For RDesign, we follow the modeling approach described in its original paper to ensure its performance is not compromised.
> > >
> > > We hope our responses have clarified your concerns. Do not hesitate to reach out if there are any remaining questions.
> > >
> > > [1] Targeting RNA structures with small molecules, Nature Reviews Drug Discovery 2022.
> > >
> > > [2] Leveraging artificial intelligence in the fight against infectious diseases, Science 2023.
> > >
> > > [3] Geometric deep learning of RNA structure, Science 2021.
> > >
> > > [4] Integrating end-to-end learning with deep geometrical potentials for ab initio RNA structure prediction, Nature Communications 2023.
> > >
> > > [5] Beyond Sequence: Impact of Geometric Context for RNA Property Prediction.

---

> > ### Author Response · Authors · 2024-11-25
> >
> > Dear Reviewer Pevo,
> >
> > We believe there may have been some misunderstandings in our previous discussion, and we have made every effort to clarify them in our latest response. We kindly ask you to reevaluate the contributions of our paper and consider revising your assessment.
> >
> > Best regards,
> >
> > The Authors

---

> > > ### Comment · Reviewer_Pevo · 2024-11-27
> > > **Thank you for your responses!**
> > >
> > > I thank the authors for their detailed responses to my questions which help clarify a lot of the issues I had with the work. I think the size generalization aspect should be clarified in the main text to avoid possible misunderstandings (similar to how the authors described in their latest response to Q1). Also I will wait for any follow-up experiments with DrFold or other candidate structure generation method. But for now, given some of my concerns are alleviated, I will change back my score to 5. I am also following the comments from other reviewers and depending on their feedback as well as any new results, I am happy to reconsider my final rating at the end of the discussion period

---

> > > > ### Author Response · Authors · 2024-11-27
> > > >
> > > > Thank you for your positive feedback on our responses! We have further clarified the aspect of size generalization in the revised version of the paper. We also appreciate your interest in the follow-up experiments, which we are actively conducting. These experiments require some time, and we will share the results with you as soon as they are available.

---

> ### Author Response · Authors · 2024-12-01
> **New experimental results**
>
> Dear Reviewer Pevo,
>
> As promised in our previous response, we are pleased to share the new experimental results, which include the use of an alternative candidate structure generator in addition to FARFAR2.
>
> Initially, we attempted to use DRfold [1]. However, we found that generating a single candidate structure using the official DRfold code took several hours, making it impractical to generate a sufficient number of candidate structures within the time constraint. As a result, we explored other recent deep learning based methods and identified the newly published RhoFold [2] as a suitable alternative. The detailed generation process is outlined below.
>
> Predicting 3D structures with RhoFold requires both 1D fasta sequences and MSA data. However, the RNAsolo dataset [3] used in our experiments does not provide the corresponding MSA data. To overcome this, we first search for similar sequences for each RNA using RNAcentral [4] and BLAST [5]. We then perform multiple sequence alignment with Infernal [6] and MAFFT [7] to generate the required MSA data files. Following the methodology outlined in the original paper [2], we randomly sample from the MSA data to ensure the uniqueness of each generated candidate structure. Finally, we compute the RMSD values between the candidate structures and their native counterparts using the pair_fit function in PyMOL, which is based on the Kabsch algorithm [8].
>
> We randomly select 20 RNAs from the training set and generate 400 candidate structures for each RNA using RhoFold, following the process described above. This results in a dataset containing 8,000 samples. We then finetune both RDesign and our EquiRNA model on this new dataset. The experimental results are presented in Table D below.
>
> **Table D: The finetuned results on the new dataset generated by RhoFold.**
> |         |          |       |       |       |         |       |       |       |
> | :--------------- | :------------- | :----- | :----- | :----- | :------- | :----- | :----- | :----- |
> |       |         |       |       |        |         |       |       |       |
> |      | Validation Set |        |        |       | Test Set |      |        |        |
> |       | ER ↓           | DR ↓   | R-ER ↓ | R-DR ↓ | ER ↓     | DR ↓   | R-ER ↓ | R-DR ↓ |
> | RDesign          | 21\.99         | 20\.79 | 1\.05  | 1\.11  | 20\.22   | 19\.34 | 0\.65  | 0\.60  |
> | RDesign+finetune | 20\.86         | 21\.02 | 0\.94  | 1\.31  | 21\.53   | 20\.45 | 0\.75  | 0\.69  |
> | EquiRNA          | 19\.77         | 19\.62 | 0\.84  | 0\.99  | **18\.22**   | **17\.79** | **0\.48**  | **0\.47**  |
> | EquiRNA+finetune | **18\.87**         | **18\.54** | **0\.76**  | **0\.88**  | 18\.78   | 18\.25 | 0\.53  | 0\.51  |
> |                  |                |        |        |        |          |        |        |        |
>
> The results show that after finetuning, our EquiRNA model achieves improved performance of all four metrics on the validation set, while RDesign shows better performance only in terms of the ER and R-ER metric. However, performance on the test set declines slightly for EquiRNA, which may be attributed to candidate structures generated entirely by FARFAR2 potentially introducing bias. Since the structures generated by RhoFold follow a different distribution from those generated by FARFAR2, the model’s performance on the test set could be partially influenced by this bias. **Notably, EquiRNA exhibits a much smaller performance drop compared to RDesign on the test set (e.g., 0.56 vs. 1.5 in terms of ER), suggesting that our model is more robust across different candidate structure generators.**
>
> Once again, thank you for your insightful suggestions, which have greatly improved our paper! We are in the process of uploading the new dataset generated by RhoFold, and you can access it via the anonymous link (https://anonymous.4open.science/r/EquiRNA-9D02). We will definitely include the new results in the revised paper and further enhance the manuscript based on your suggestions. We hope this response adequately addresses your concerns.
>
>
> [1] Li Y et al. Integrating end-to-end learning with deep geometrical potentials for ab initio RNA structure prediction. Nature Communications, 2023.
>
> [2] Shen T et al. Accurate RNA 3D structure prediction using a language model-based deep learning approach. Nature Methods, 2024.
>
> [3] Adamczyk B et al. RNAsolo: a repository of cleaned PDB-derived RNA 3D structures. Bioinformatics, 2022.
>
> [4] https://rnacentral.org/
>
> [5] https://blast.ncbi.nlm.nih.gov/Blast.cgi
>
> [6] Nawrocki E P et al. Infernal 1.1: 100-fold faster RNA homology searches. Bioinformatics, 2013.
>
> [7] Katoh K et al. MAFFT: a novel method for rapid multiple sequence alignment based on fast Fourier transform. Nucleic acids research, 2002.
>
> [8] Kabsch W. A solution for the best rotation to relate two sets of vectors. Acta Crystallographica Section A: Crystal Physics, Diffraction, Theoretical and General Crystallography, 1976.
>
> Best regards,
>
> The Authors

---

> ### Author Response · Authors · 2024-12-03
> **Looking forward to your responses**
>
> Dear Reviewer Pevo,
>
> As the rebuttal phase is nearing its final hours, we are eagerly awaiting your feedback! We have made every effort to thoroughly address each of your concerns and would like to know if our responses meet your expectations. We would be extremely grateful if you could reconsider our paper and consider raising your score!
>
> Thank you once again for your hard work and invaluable feedback during the review process!
>
> Best regards,
>
> The Authors

---

> ### Comment · Reviewer_Pevo · 2024-12-03
>
> I thank the authors for doing additional experiments in response to my question. Please include this analysis in the paper. I am raising my score to 6

---

> > ### Author Response · Authors · 2024-12-03
> >
> > We are delighted to see that you recognize our response! We will incorporate the relevant analysis into our paper. Thank you again for your valuable feedback and effort during the review process!

---

### Official Review · Reviewer_uusj · 2024-11-04

**Soundness:** 3
**Presentation:** 3
**Contribution:** 2
**Rating:** 6
**Confidence:** 4

**Summary:**

The paper proposes EquiRNA, a hierarchical equivariant neural network designed to evaluate RNA 3D structures and generalize across varying RNA sizes, along with a new benchmark dataset rRNAsolo. The key strength is its practical approach to scaling equivariant neural networks for RNA data through multi-level equivariant message passing and size-insensitive k-NN sampling. Additionally, the paper introduces a new curated RNA dataset for structure fitness prediction task which is an important contribution given the scarcity of RNA datasets. However, the choice of E(3) over SE(3) equivariance ignores molecular chirality, the subunit and nucleotide level message-passing mechanisms lack clear motivation, and the empirical validation would benefit from multiple dataset splits given the relatively small number of samples in the datasets.

**Strengths:**

1. The paper tackles important practical limitations of the scalability of equivariant neural networks applied to real biological data.
2. The paper focuses on RNA which is a much less saturated domain compared to other biomolecular modalities such as proteins. Extending a machine learning toolkit for RNA analysis is an important contribution.
3. The paper introduces a new dataset rRNAsolo. Authors make great effort in collecting the data and purifying it on multiple levels. This is an important contribution given the scarcity of RNA datasets for machine learning.
4. The subunit-level message passing is novel and the ablation study underlines its importance for the fitness prediction task.
5. The k-NN sampling strategy is a simple approach for that potentially robustifies a model against an input size distribution shift.

**Weaknesses:**

1. The method focuses on E(3)-equivariance while in reality SE(3)-equivariance is the most relevant due to the molecular chirality. The E(3)-invariant/equivariant method does not discriminate between RNA molecules and their reflection which often comes with drastically different properties. This severely limits the practical applicability of the proposed method. Authors may consider the method presented in [1] (Eq.6-7) as a relatively easy way to break the symmetry to reflections and achieve SE(3)-equivariance.
2. The size generalization problem requires further elaboration. RNAs of different lengths usually correspond to different types, and hence different functions and structures. With this, it is hard to see how a network could generalize to another type of RNA with possibly completely different biological function/structure without observing it during training.
3. The subunit-level message passing (lines 273-300) is not explained clearly, and the particular form of message-passing in equation (2) is not well-motivated. Without this clear motivation, I fail to see why this form of message-passing will bring any advantage compared to a simple EGNN-like message-passing scheme.
4. Similarly, the particular form of nucleotide-level message passing is not well-motivated. The framework of learnable templates and matrix-valued message function in Eq. 3 appears as unnecessary over-complication. I am wondering why the simplest approach of averaging the nucleotide coordinates (similar to how it is done with invariant nucleotide features) is not enough here. At least, I would expect an ablation study with respect to simple averaging.
5. The experiments are performed only on one split of the datasets. Given the relatively small number of samples in the datasets, it is good to run the methods over several splits and provide mean and standard deviation results. Further, the benchmarking on the proposed rRNAsolo dataset will be much more reliable if the dataset comes with several provided train\val\test splits. This will significantly increase the weight of the rRNAsolo contribution.
6. The related work on Geometric GNNs misses elaboration on recent scalable equivariance methods: [2,3,4], etc. This is important since the paper claims to tackle the practical limitations of the scalability of equivariant neural networks

[1]. A. Schneuing et al. Structure-based drug design with equivariant diffusion models.
[2]. A. Moskalev et al. SE(3)-Hyena Operator for Scalable Equivariant Learning. GRaM@ICML 2024. \
[3]. F. Sestak et al. VN-EGNN: Equivariant Graph Neural Networks with Virtual Nodes Enhance Protein Binding Site Identification. ICML 2024. \
[4]. E. Bekkers et al. Fast, Expressive SE(n) Equivariant Networks through Weight-Sharing in Position-Orientation Space. ICLR 2024.

Minor:
- line 346 typo: size-generation -> size-generalization

**Questions:**

I suggest authors address weaknesses for the rebuttal.

---

> ### Author Response · Authors · 2024-11-22
> **Responses to Reviewer uusj (Part 1/3)**
>
> We greatly appreciate your constructive feedback! We provide detailed responses to your questions below and have made corresponding revisions to our paper.
>
> > **Q1: Authors may consider the method presented in [1] (Eq.6-7) as a relatively easy way to break the symmetry to reflections and achieve SE(3)-equivariance.**
>
> **A1:** Nice suggestion! Thank you very much for pointing this out. We attempt the approach from [1], by incorporating Eq. 6-7 of [1] into our model to further account for molecular chirality. The results are shown in the table below. As can be seen, including chirality actually led to a slight performance drop.
> This is quite reasonable. In fact, our task is to predict the RMSD between candidate RNA structures and stable structures, which is an E(3)-invariant quantity and is insensitive to structural flipping. Therefore, for our task, an E(3)-invariant model is more appropriate than an SE(3)-invariant model.
> Of course, we agree that if the task involves predicting molecular properties sensitive to chirality, such as optical activity [2], then an SE(3)-invariant model would indeed be more suitable. We have further clarified this in the revised version of the paper.
>
> **Table A: The results of rRNAsolo when considering molecular chirality.**
>
> |                  | Validation Set | Test Set |        |        |       |       |        |        |
> | ---------------- | -------------- | -------- | ------ | ------ | ----- | ----- | ------ | ------ |
> |                  | ER ↓           | DR  ↓    | R-ER ↓ | R-DR ↓ | ER ↓  | DR  ↓ | R-ER ↓ | R-DR ↓ |
> | EquiRNA with [1] | 20.66          | 21.02    | 0.92   | 1.13   | 18.85 | 19.28 | 0.53   | 0.60   |
> | EquiRNA          | **19.77**          | **19.62**    | **0.84**   | **0.99**   | **18.22** | **17.79** | **0.48**   | **0.47**   |
>
> > **Q2: The size generalization problem requires further elaboration. RNAs of different lengths usually correspond to different types, and hence different functions and structures.**
>
> **A2:** Thank you for your insightful comment! We agree that RNAs of varying lengths often correspond to distinct types, functions, or structures. Nonetheless, we believe that generalization across RNAs of different lengths is achievable through the use of our proposed hierarchical model. While short and long RNAs may differ in their functions and structures, they share common substructures and local patterns, such as nucleotides. Our model generalizes across different RNA sizes by capturing these shared features and maximizing the reuse of common building blocks (i.e., nucleotides). Additionally, long RNAs are often composed of smaller functional modules, whose characteristics can be found in short RNAs. Therefore, by training the model on short RNAs, we effectively extract modular features that are applicable to long RNAs, enabling robust generalization. This approach is supported by prior work, such as FARFAR2 [3], RNAComposer [4], and 3dRNA [5], which have demonstrated the effectiveness of fragment-based decomposition of RNAs.
>
> > **Q3: The subunit-level message passing (lines 273-300) is not explained clearly, and the particular form of message-passing in equation (2) is not well-motivated. I fail to see why this form of message-passing will bring any advantage compared to a simple EGNN-like message-passing scheme.**
>
> **A3:** Thank you for your thoughtful comment! We are happy to provide additional clarification regarding the subunit-level message passing.
> From a biological perspective, the interactions between the phosphate-ribose backbone and nucleobases are highly complex, characterized by significant heterogeneity and dynamic behavior rather than simple linear relationships. To effectively capture this complexity and the varying importance of these interactions, we employ attention-based methods to learn adaptive weights ($\alpha_{ij}$). This approach allows our model to better reflect the nuanced and context-dependent nature of these molecular interactions.
> To further validate the advantages of our design, we additionally conduct an experiment by replacing the attention mechanism with a simpler EGNN-like message-passing scheme, as you suggested. The results, presented in Table B, show that the EGNN-like scheme performs worse compared to our original strategy. These findings highlight the effectiveness of the attention mechanism in capturing the intricacies of subunit-level interactions.

---

> ### Author Response · Authors · 2024-11-22
> **Responses to Reviewer uusj (Part 2/3)**
>
> > **Q4: Similarly, the particular form of nucleotide-level message passing is not well-motivated. The framework of learnable templates and matrix-valued message function in Eq. 3 appears as unnecessary over-complication. At least, I would expect an ablation study with respect to simple averaging.**
>
> **A4:** Nice suggestion! In the nucleotide-level module, we aim to model atom-level interactions between nucleotides to capture finer-grained features. Directly averaging all atomic coordinates may cause the model to lose full-atom information during the message passing in nucleotide-level module. To validate this hypothesis, we follow your suggestion by averaging the atomic coordinates within each nucleotide. The experimental results, as shown in Table B, indicate that this approach adversely affects the model's performance.
>
> **Table B:  Ablation studies on rRNAsolo.**
>
> |                               | ER ↓  | DR  ↓ | R-ER | R-DR |
> | ----------------------------- | ----- | ----- | ---- | ---- |
> | EGNN-like at Subunit-Level    | 18.86 | 19.75 | 0.53 | 0.64 |
> | Averaging at Nucleotide-Level | 19.87 | 18.66 | 0.62 | 0.55 |
> | EquiRNA                       | **18.22** | **17.79** | **0.48** | **0.47**|
>
> > **Q5: Given the relatively small number of samples in the datasets, it is good to run the methods over several splits and provide mean and standard deviation results. Further, the benchmarking on the proposed rRNAsolo dataset will be much more reliable if the dataset comes with several provided train\val\test splits.**
>
> **A5:**
> Thank you for your valuable suggestion! In our rRNAsolo dataset, the training set comprises 200 RNAs, each generating approximately 400 candidate structures, resulting in about 80,000 samples in total. This constitutes a reasonably large training set. The number of RNAs is limited by the availability of 3D RNA structures, and the 200 RNAs included represent the maximum we could obtain following a rigorous screening process. Details of this process are provided in Section 3.2 of the main paper and Appendix C.
>
> As you rightly pointed out, training on multiple splits would enhance the credibility of our results. However, our study focuses specifically on size generalization, where the lengths of training RNAs are intentionally smaller than those of the test RNAs. Randomly shuffling the training, validation, and test sets would disrupt this setup and compromise our goal of evaluating size-based generalization.
>
> Nonetheless, to address your suggestion and further validate the robustness of our model, we extend our study to a more general scenario where size generalization is not a constraint. Based on the clustering approach described in Appendix C.1, we randomly split the training, validation, and test sets at the cluster level. Due to time and resource limitations, we test three random split configurations, with each split containing 20k, 6k, and 6k samples in the training, validation, and test sets, respectively. The results, presented in Appendix J of the revised version, demonstrate that our model consistently achieves superior performance across nearly all three splits, further underscoring its robustness and effectiveness. For your reference, Appendix J also lists the PDB IDs of the RNAs included in each split.

---

> ### Author Response · Authors · 2024-11-22
> **Responses to Reviewer uusj (Part 3/3)**
>
> > **Q6: The related work on Geometric GNNs misses elaboration on recent scalable equivariance methods: [2,3,4], etc. This is important since the paper claims to tackle the practical limitations of the scalability of equivariant neural networks.**
>
> **A6:** Thank you very much for bringing these papers to our attention. We have cited and discussed them in the revised version of our manuscript. Specifically, SE(3)-Hyena [6], VN-EGNN [7], and Ponita [8] employ global context (the SE(3)-Hyena operator, virtual nodes, and global group actions, respectively) to enhance the scalability of equivariant neural networks. These approaches represent excellent strategies for further improving general-purpose models.
>
> In contrast, our EquiRNA is a specialized design tailored to leverage the prior knowledge of RNA nucleotide structures, enabling the effective application of equivariant neural networks to large RNA molecules. This specialization allows EquiRNA to address the unique challenges of modeling RNA, distinguishing it from the more general approaches discussed.
>
> > **Q7: typo: size-generation -> size-generalization**
>
> **A7:** Thank you for your suggestion! We have corrected the typo.
>
> [1] A. Schneuing et al. Structure-based drug design with equivariant diffusion models.
>
> [2] https://en.wikipedia.org/wiki/Chirality_(chemistry).
>
> [3] Watkins A M et al. FARFAR2: Improved De Novo Rosetta Prediction of Complex Global RNA Folds. Structure 2020.
>
> [4] Biesiada M et al. RNAComposer and RNA 3D structure prediction for nanotechnology. Methods 2016.
>
> [5] Zhang Y et al. 3dRNA: 3D Structure Prediction from Linear to Circular RNAs. J. Mol. Biol. 2022.
>
> [6] A. Moskalev et al. SE(3)-Hyena Operator for Scalable Equivariant Learning. GRaM@ICML 2024.
>
> [7] F. Sestak et al. VN-EGNN: Equivariant Graph Neural Networks with Virtual Nodes Enhance Protein Binding Site Identification. ICML 2024.
>
> [8] E. Bekkers et al. Fast, Expressive SE(n) Equivariant Networks through Weight-Sharing in Position-Orientation Space. ICLR 2024.

---

> ### Author Response · Authors · 2024-11-27
>
> Dear Reviewer uusj,
>
> We sincerely appreciate the valuable time you have dedicated to reviewing our paper! We have carefully addressed all of your questions and concerns, and we kindly seek your assessment of our responses. We would be tremendously grateful if you would consider raising your score!
>
> Best regards,
>
> The Authors

---

> ### Author Response · Authors · 2024-12-01
> **Looking forward to your responses**
>
> Dear Reviewer uusj,
>
> With the rebuttal deadline drawing near, we earnestly await your feedback. We would like to know if you are satisfied with our responses and willing to reevaluate our paper. We would greatly appreciate it if you could consider raising your score!
>
> Thank you again for reviewing our paper and providing valuable feedback, which is crucial for improving the quality of our work!
>
> Best regards,
>
> The Authors

---

> ### Author Response · Authors · 2024-12-03
> **Looking forward to your responses**
>
> Dear Reviewer uusj,
>
> As the rebuttal phase is nearing its final hours, we are eagerly awaiting your feedback! We have made every effort to thoroughly address each of your concerns and would like to know if our responses meet your expectations. We would be extremely grateful if you could reconsider our paper and consider raising your score!
>
> Thank you once again for your hard work and invaluable feedback during the review process!
>
> Best regards,
>
> The Authors

---

> > ### Comment · Reviewer_uusj · 2024-12-03
> > **Reviewer response**
> >
> > I appreciate the authors' effort in addressing my concerns. While I do not believe that Q2 regarding size generalizability has been adequately addressed (and I am skeptical that it can be at all), I am convinced that the additional experiments and ablations provided by the authors during the rebuttal significantly strengthen the paper. Overall, I am convinced that the strengths (marginally) outweigh the weaknesses, and I thus raise my score from 5 to 6.

---

> > > ### Author Response · Authors · 2024-12-03
> > >
> > > Thank you for your positive feedback on our rebuttal! Regarding Q2, as you mentioned, we have indeed only made preliminary attempts to address the issue of size generalization. Fully resolving this challenge is indeed very difficult, and we will continue to explore this issue in the future. Thank you again for your valuable feedback and effort during the review process!

---

### Author Response · Authors · 2024-12-04
**General Response**

We genuinely thank all reviewers and ACs for their time and efforts on reviewing the paper. We are very glad that the reviewers acknowledged the problems we studied, the dataset we proposed, and the models we developed, and their feedback really gave us a lot of inspiration and enlightenment.

Our new dataset, rRNAsolo, offers a more comprehensive benchmark compared to existing datasets, providing a reliable foundation for the preliminary exploration of size generalization in RNA structure evaluation tasks. Additionally, our proposed EquiRNA model acts as a promising approach for the initial exploration of tackling the size generalization problem. We hope that these contributions will garner increased attention to the RNA research field and foster its development. Any feedback provided during the rebuttal process will be thoroughly considered and integrated into the final version.

We have published the pipeline used for dataset construction, inference code, and test datasets via an anonymous link: https://anonymous.4open.science/r/EquiRNA-9D02. Moreover, to address the reviewers' concerns, we have added additional experiments and figures as follows.

**Table 9** in Appendix G reports the complexity analysis of various methods.

**Table 10** in Appendix H shows the results of rRNAsolo when considering molecular chirality.

**Table 11** in Appendix I shows the ablation studies at subunit-Level and nucleotide-Level on rRNAsolo.

**Table 12-14** in Appendix J reports the results on three random split configurations where size generalization is not a constraint.

**Table 15** in Appendix L compares EquiRNA with more leading models (ProNet and Equiformer).

**Figure 6** in Appendix M shows some error cases.

**Table 16** in Appendix N reports the results of EquiRNA on rRNAsolo when considering noise.

**Figure 7** in Appendix O illustrates the graph structures in Section 3.1 more clearly.

**Table D** in the responses to Reviewer Pevo/Table B in the responses to Reviewer egW8 shows the finetuned results of EquiRNA and RDeisgn on the new dataset generated by RhoFold.

---

### Author Response · Authors · 2024-12-04
**Rebuttal Acknowledgment**

Dear Reviewers and Area Chairs,

We would like to express our sincere appreciation for your diligent efforts, valuable feedback, and constructive suggestions. Your encouraging remarks and considerate suggestions are genuinely praiseworthy.

Through our discussions and the reviewers' responses, we believe we have addressed the major concerns raised by all reviewers. Among the rebuttal, 3 out of 4 reviewers (Reviewer uusj, Reviewer Pevo, and Reviewer egW8) raised their ratings. Specifically, both Reviewer uusj and Reviewer Pevo raised from 5 to 6, and Reviewer egW8 raised from 6 to 8. Now all the reviewers provided us with positive judgement. This outcome has greatly benefited and encouraged us, and we would like to thank all of you for your valuable support!

Once again, we express our sincere gratitude for your time and dedication in reviewing our work. Your valuable input has greatly enhanced the quality of our manuscript. Thank you!

Best regards,

The Authors

---

### Meta-Review · Area_Chair_D2sJ · 2024-12-20

**Metareview:**

This submission received feedback from four reviewers, all of whom unanimously provided positive reviews. Notably, the additional experiments and explanations provided by the authors in the rebuttal led three reviewers to raise their original scores, e.g., Reviewer uusj and Reviewer Pevo raised their scores from 5 to 6, and Reviewer egW8 raised theirs from 6 to 8. Therefore, my recommendation is 'Accept (poster).'”

**Additional Comments On Reviewer Discussion:**

This submission received feedback from four reviewers, all of whom unanimously provided positive reviews. Notably, the additional experiments and explanations provided by the authors in the rebuttal led three reviewers to raise their original scores, e.g., Reviewer uusj and Reviewer Pevo raised their scores from 5 to 6, and Reviewer egW8 raised theirs from 6 to 8.

---

### Decision · Program_Chairs · 2025-01-22

Accept (Poster)